# Evaluation of Rust code verbosity, understandability and complexity

Luca Ardito[1], Luca Barbato[2], Riccardo Coppola[1] and Michele Valsesia[1]

[1] Department of Control and Computer Engineering, Polytechnic Institute of Turin, Torino, Piemonte, Italia
[2] Luminem, Torino, Piemonte, Italia



## ABSTRACT

Rust is an innovative programming language initially implemented by Mozilla, developed to ensure high performance, reliability, and productivity. The final purpose of this study consists of applying a set of common static software metrics to programs written in Rust to assess the verbosity, understandability, organization, complexity, and maintainability of the language. To that extent, nine different implementations of algorithms available in different languages were selected.
We computed a set of metrics for Rust, comparing them with the ones obtained from C and a set of object-oriented languages: C++, Python, JavaScript, TypeScript.
To parse the software artifacts and compute the metrics, it was leveraged a tool called *rust-code-analysis* that was extended with a software module, written in Python, with the aim of uniforming and comparing the results. The Rust code had an average verbosity in terms of the raw size of the code. It exposed the most structured source organization in terms of the number of methods. Rust code had a better Cyclomatic Complexity, Halstead Metrics, and Maintainability Indexes than C and C++ but performed worse than the other considered object-oriented languages. Lastly, the Rust code exhibited the lowest COGNITIVE complexity of all languages. The collected measures prove that the Rust language has average complexity and maintainability compared to a set of popular languages. It is more easily maintainable and less complex than the C and C++ languages, which can be considered syntactically similar. These results, paired with the memory safety and safe concurrency characteristics of the language, can encourage wider adoption of the language of Rust in substitution of the C language in both the open-source and industrial environments.

# INTRODUCTION

Software maintainability is defined as the ease of maintaining software during the delivery of its releases. Maintainability is defined by the ISO 9126 standard as "The ability to identify and fix a fault within a software component" (*ISO, 1991*), and by the ISO/IEC 25010:2011 standard as "degree of effectiveness and efficiency with which a product or system can be modified by the intended maintainers" (*ISO/IEC, 2011*). Maintainability is an integrated software measure that encompasses some code characteristics, such as readability, documentation quality, simplicity, and understandability of source code (*Aggarwal, Singh & Chhabra, 2002*).

Corresponding author
Luca Ardito, luca.ardito@polito.it

Maintainability is a crucial factor in the economic success of software products. It is commonly accepted in the literature that the most considerable cost associated with any software product over its lifetime is the maintenance cost (*Zhou & Leung, 2007*). The maintenance cost is influenced by many different factors, for example, the necessity for code fixing, code enhancements, the addition of new features, poor code quality, and subsequent need for refactoring operations (*Nair & Swaminathan, 2020*).

Hence, many methodologies have consolidated in software engineering research and practice to enhance this property. Many metrics have been defined to provide a quantifiable and comparable measurement for it (*Nuñez-Varela et al., 2017*). Many metrics measure lower-level properties of code (e.g., related to the number of lines of code and code organization) as proxies for maintainability. Several comprehensive categorizations and classifications of the maintainability metrics presented in the literature during the last decades have been provided, for example, the one by *Frantz et al. (2019)* provides a categorization of 25 different software metrics under the categories of *Size*, *Coupling*, *Complexity*, and *Inheritance*.

The academic and industrial practice has also provided multiple examples of tools that can automatically compute software metrics on source code artifacts developed in many different languages (*Mshelia, Apeh & Edoghogho, 2017*). Several frameworks have also been described in the literature that leverage combinations of software code metrics to predict or infer the maintainability of a project (*Kaur, Kaur & Pathak, 2014b*; *Amara & Rabai, 2017*; *Mshelia & Apeh, 2019*). The most recent work in the field of metric computation is aiming at applying machine learning-based approaches to the prediction of maintainability by leveraging the measurements provided by static analysis tools (*Schnappinger et al., 2019*).

However, the benefit of the massive availability of metrics and tooling for their computation is contrasted by the constant emergence of novel programming languages in the software development community. In most cases, the metrics have to be readapted to take into account newly defined syntaxes, and existing metric-computing tools cannot work on new languages due to the unavailability of parsers and metric extraction modules. For recently developed languages, the unavailability of appropriate tooling represents an obstacle for empirical evaluations on the maintainability of the code developed using them.

This work provides a first evaluation of verbosity, code organization, understandability, and complexity of Rust, a newly emerged programming language similar in characteristics to C++, developed with the premises of providing better maintainability, memory safety, and performance (*Matsakis & Klock, 2014*). To this purpose, we (i) adopted and extended a tool to compute maintainability metrics that support this language; (ii) developed a set of scripts to arrange the computed metrics into a comparable JSON format; (iii) executed a small-scale experiment by computing static metrics for a set of programming languages, including Rust, analyzing and comparing the final results. To the best of our knowledge, no existing study in the literature has provided computations of such metrics for the Rust language and the relative comparisons with other languages.

The remainder of the manuscript is structured as follows: "Background and Related Work" provides background information about the Rust language and presents a brief review of state-of-the-art tools available in the literature for the computation of metrics related to maintainability; "Study Design" describes the methodology used to conduct our experiment, along with a description of the developed tools and scripts, the experimental subjects used for the evaluation, and the threats to the validity of the study; "Results and Discussion" presents and discusses the collected metrics; "Conclusion and Future Work" concludes the paper by listing its main findings and providing possible future directions of this study.

## BACKGROUND AND RELATED WORK

This section provides background information about the Rust language characteristics, studies in the literature that analyzes its advantages, and the list of available tools present in the literature to measure metrics used as a proxy to quantify software projects' maintainability.

### The Rust programming language

Rust is an innovative programming language initially developed by Mozilla and is currently maintained and improved by the Rust Foundation (https://www.rust-lang.org/).

The main goals of the Rust programming language are: memory-efficiency, with the abolition of garbage collection, with the final aim of empowering performance-critical services running on embedded devices, and easy integration with other languages; reliability, with a rich type system and ownership model to guarantee memory-safety and thread-safety; productivity, with an integrated package manager and build tools.

Rust is compatible with multiple architectures and is quite pervasive in the industrial world. Many companies are currently using Rust in production today for fast, low-resource, cross-platform solutions: for example, software like Firefox, Dropbox, and Cloudflare use *Rust (2020)*.

The Rust language has been analyzed and adopted in many recent studies from academic literature. *Uzlu & Şaykol (2017)* pointed out the appropriateness of using Rust in the Internet of Things domain, mentioning its memory safety and compile-time abstraction as crucial peculiarities for the usage in such domain. *Balasubramanian et al. (2017)* show that Rust enables system programmers to implement robust security and reliability mechanisms more efficiently than other conventional languages. *Astrauskas et al. (2019)* leveraged Rust's type system to create a tool to specify and validate system software written in Rust. *Köster (2016)* mentioned the speed and high-level syntax as the principal reasons for writing in the Rust language the Rust-Bio library, a set of safe bioinformatic algorithms. *Levy et al. (2017)* reported the process of developing an entire kernel in Rust, with a focus on resource efficiency. These common usages of Rust in such low-level applications encourage thorough analyses of the quality and complexity of a code with Rust.
**Table 1 Languages supported by the metrics tools.**

| Language | CBR insight | CCFinderX | CKJM | CodeAnalyzers | Halstead metrics tool | Metrics reloaded | Squale |
|---|---|---|---|---|---|---|---|
| C | x | x | | x | | x | x |
| C++ | x | x | | x | x | | x |
| C# | x | x | | x | | | |
| Cobol | x | x | | x | | | x |
| Java | x | x | x | x | x | x | |
| Rust | | | | | | | |
| Others | x | | | x | | | |

## Tools for measuring static code quality metrics

Several tools have been presented in academic works or are commonly used by practitioners to measure quality metrics related to maintainability for software written in different languages.

In our previous works, we conducted a systematic literature review that led us to identify fourteen different open-source tools that can be used to compute a large set of different static metrics (*Ardito et al., 2020b*). In the review, it is found that the following set of open-source tools can cover most of quality metrics defined in the literature, for the most common programming languages: *CBR Insight*, a tool based on the closed-source metrics computation Understand framework, that aims at computing reliability and maintainability metrics (*Ludwig & Cline, 2019*); *CCFinderX*, a tool tailored for finding duplicate code fragments (*Matsushita & Sasano, 2017*); *CKJM*, a tool to compute the C&K metrics suite and method-related metrics for Java code (*Kaur, Kaur & Pathak, 2014a*); *CodeAnalyzers*, a tool supporting more than 25 software maintainability metrics, that covers the highest number of programming languages along with CBR Insight (*Sarwar et al., 2008*); *Halstead Metrics Tool*, a tool specifically developed for the computation of the Halstead Suite (*Hariprasad et al., 2017*); *Metrics Reloaded*, able to compute many software metrics for C and Java code either in a plug-in for IntelliJ IDEA or through command line (*Saifan, Alsghaier & Alkhateeb, 2018*); *Squale*, a tool to measure high-level quality factors for software and measuring a set of code-level metrics to predict economic aspects of software quality (*Ludwig, Xu & Webber, 2017*).

Table 1 reports the principal programming languages supported by the described tools. For the sake of conciseness, only the languages that were supported by at least two of the tools are reported. With this comparison, it can be found that none of the considered tools is capable of providing metric computation facilities for the Rust language.

As additional limitations of the identified set of tools, it can be seen that the tools do not provide complete coverage of the most common metrics for all the tools (e.g., the Halstead Metric suite is computed only by the Halstead Metrics tool), and in some cases, (e.g., CodeAnalyzer), the number of metrics is limited by the type of acquired license. Also, some of the tools (e.g., Squale) appear to have been discontinued by the time of the writing of this article.

| Table 2 Case study definition template (*Robson & McCartan, 2016*). | |
|---|---|
| **Objective** | **Evaluation of code verbosity, understandability and complexity** |
| The case | Development with the Rust programming language |
| Theory | Static measures for software artifacts |
| Research questions | What is the verbosity, organization, complexity and maintainability of Rust? |
| Methods | Comparison of Rust static measurements with other programming languages |
| Selection strategy | Open-source multi-language repositories |

## STUDY DESIGN

This section reports the goal, research questions, metrics, and procedures adopted for the conducted study.

To report the plan for the experiment, the template defined by Robson was adopted (*Robson & McCartan, 2016*). The purpose of the research, according to Robson's classification, is *Exploratory*, that is, to find out whats is happening, seeking new insights, and generating ideas and hypotheses for future research. The main concepts of the definition of the study are reported in Table 2.

In the following subsections, the best practices for case study research provided by Runeson and Host are adopted to organize the presentation of the study (*Jedlitschka & Pfahl, 2005*). More specifically, the following elements are reported: goals, research questions, and variables; objects; instrumentation; data collection and analysis procedure; evaluation of validity.

### Goals, research questions and variables

The high-level goal of the study can be expressed as:

*Analyze and evaluate the characteristics of the Rust programming language, focusing on verbosity, understandability and complexity measurements, measured in the context of open-source code, and interpreting the results from developers and researchers' standpoint.*

Based on the goal, the research questions that guided the definition of the experiment are obtained. Four different aspects that deserve to be analyzed for code written in Rust programming language were identified, and a distinct Research Question was formulated for each of them. In the following, the research questions are listed, along with a brief description of the metrics adopted to answer them. Table 3 reports a summary of all the metrics.

The comparisons between different programming languages were made through the use of static metrics. A static metric (opposed to dynamic or runtime metrics) is obtained by parsing and extracting information from a source file without depending on any information deduced at runtime.

- **RQ1**: What is the verbosity of Rust code with respect to code written in other programming languages?

**Table 3 List of metrics used in this study.**

| RQ | Acronym | Name | Description |
|---|---|---|---|
| RQ1 | SLOC | Source Lines of Code | It returns the total number of lines in a file |
| | PLOC | Physical Lines of Code | It returns the total number of instructions and comment lines in a file |
| | LLOC | Logical Lines of Code | It returns the number of logical lines (statements) in a file |
| | CLOC | Comment Lines of Code | It returns the number of comment lines in a file |
| | BLANK | Blank Lines of Code | Number of blank statements in a file |
| RQ2 | NOM | Number of Methods | It returns the number of methods in a source file |
| | NARGS | Number of Arguments | It counts the number of arguments for each method in a file |
| | NEXITS | Number of Exit Points | It counts the number of exit points of each method in a file |
| RQ3 | CC | McCabe's Cyclomatic Complexity | It calculates the code complexity examining the control flow of a program; the original McCabe's definition of cyclomatic complexity is the the maximum number of linearly independent circuits in a program control graph (*Gill & Kemerer, 1991*) |
| | COGNITIVE | Cognitive Complexity | It is a measure of how difficult a unit of code is to intuitively understand, by examining the cognitive weights of basic software control structures (*Jingqiu & Yingxu, 2003*) |
| | Halstead | Halstead suite | A suite of quantitative intermediate measures that are translated to estimations of software tangible properties, for example, volume, difficulty and effort (see Table 4 for details) |
| RQ4 | MI | Maintainability Index | A composite metric that incorporates a number of traditional source code metrics into a single number that indicates relative maintainability (see Table 5 for details about the considered variants) (*Welker, 2001*) |

To answer RQ1, the size of code artifacts written in Rust was measured in terms of the number of code lines in a source file. Four different metrics have been defined to differentiate between the nature of the inspected lines of code:

- SLOC, i.e., Source lines of code;
- CLOC, Comment Lines of Code;
- PLOC, Physical Lines of Code, including both the previous ones;
- LLOC, Logical Lines of Code, returning the count of the statements in a file;
- BLANK, Blank Lines of Code, returning the number of blank lines in a code.

The rationale behind using multiple measurements for the lines of code can be motivated by the need for measuring different facets of the size of code artifacts and of the relevance and content of the lines of code. The measurement of physical lines of code (PLOC) does not take into consideration blank lines or comments; the count, however, depends on the physical format of the statements and programming style since multiple PLOC can concur to form a single logical statement of the source code. PLOC are sensitive to logically irrelevant formatting and style conventions, while LLOC are less sensitive to these aspects (*Nguyen et al., 2007*). In addition to that, the CLOC and BLANK measurements allow a finer analysis of the amount of documentation (in terms of used APIs and explanation of complex parts of algorithms) and formatting of a source file.

- **RQ2**: How is Rust code organized with respect to code written in other programming languages?

**Table 4 The halstead metrics suite.**

| Measure | Symbol | Formula |
|---|---|---|
| Base measures | η1 | Number of distinct operators |
| | η2 | Number of distinct operands |
| | N1 | Total number of occurrences of operators |
| | N2 | Total number of occurrences of operands |
| Program length | N | $N = N1 + N2$ |
| Program vocabulary | η | $\eta = \eta1 + \eta2$ |
| Volume | V | $V = N * \log_2(\eta)$ |
| Difficulty | D | $D = \eta1/2 * N2/\eta2$ |
| Program level | L | $L = 1/D$ |
| Effort | E | $E = D * V$ |
| Estimated program length | H | $H = \eta1 * \log_2(\eta1) + \eta2 * \log_2(\eta2)$ |
| Time required to program (in seconds) | T | $T = E/18$ |
| Number of delivered bugs | B | $B = E^{2/3}/3000$ |
| Purity ratio | PR | $PR = H/N$ |

To answer RQ2, the source code structure was analyzed in terms of the properties and functions of source files. To that end, three metrics were adopted: NOM, Number of Methods; NARGS, Number of Arguments; NEXITS, Number of exits. NARGS and NEXITS are two software metrics defined by Mozilla and have no equivalent in the literature about source code organization and quality metrics. The two metrics are intuitively linked with the easiness in reading and interpreting source code: a function with a high number of arguments can be more complex to analyze because of a higher number of possible paths; a function with many exits may include higher complexity in reading the code for performing maintenance efforts.

- **RQ3**: What is the complexity of Rust code with respect to code written in other programming languages?

To answer RQ3, three metrics were adopted: CC, McCabe's Cyclomatic Complexity; COGNITIVE, Cognitive Complexity; and the *Halstead suite*. The Halstead Suite, a set of quantitative complexity measures originally defined by Maurice Halstead, is one of the most popular static code metrics available in the literature (*Hariprasad et al., 2017*). Table 4 reports the details about the computation of all operands and operators. The metrics in this category are more high-level than the previous ones and are based on the computation of previously defined metrics as operands.

- **RQ4:** What are the composite maintainability indexes for Rust code with respect to code written in other programming languages?

To answer RQ4, the Maintainability Index was adopted, that is, a composite metric originally defined by *Oman & Hagemeister (1992)* to provide a single index of maintainability for software. Three different versions of the Maintainability Index are considered. First, the original version by *Oman & Hagemeister (1992)*. Secondly, the

**Table 5 Considered variants of the MI metric.**

| Acronym | Meaning | Formula |
|---------|---------|---------|
| $MI_O$ | Original maintainability index | $171.0 - 5.2 * \ln(V) - 0.23 * CC - 16.2 * \ln(SLOC)$ |
| $MI_{SEI}$ | MI by Software Engineering Institute | $171.0 - 5.2 * \log_2(V) - 0.23 * CC - 16.2 * \log_2(SLOC) + 50.0 * \sin\left(\sqrt{2.4 * (CLOC/SLOC)}\right)$ |
| $MI_{VS}$ | MI implemented in Visual Studio | $\max(0, (171 - 5.2 * \ln(V) - 0.23 * CC - 16.2 * \ln(SLOC)) * 100/171)$ |

**Table 6 Maintainability ranges of source code according to different formulas for the MI metric.**

| Variant | Low maintainability | Medium maintainability | High maintainability |
|---------|---------------------|------------------------|----------------------|
| Original | MI < 65 | 65 < MI < 85 | MI > 85 |
| SEI | MI < 65 | 65 < MI < 85 | MI > 85 |
| VS | MI < 10 | 10 < MI < 20 | MI > 20 |

version defined by the Software Engineering Institute (SEI), originally promoted in the C4 Software Technology Reference Guide (*Bray et al., 1997*); the SEI adds to the original formula a specific treatment for the comments in the source code (i.e., the CLOC metric), and it is deemed by research as more appropriate given that the comments in the source code can be considered correct and appropriate (*Welker, 2001*). Finally, the version of the MI metric implemented in the Visual Studio IDE (*Microsoft, 2011*); this formula resettles the MI value in the 0–100 range, without taking into account the distinction between CLOC and SLOC operated by the SEI formula (*Molnar & Motogna, 2017*).

The respective formulas are reported in Table 5. The interpretation of the measured MI varies according to the adopted formula to compute it: the ranges for each of them are reported in Table 6. For the traditional and the SEI formulas of the MI, a value over 85 indicates easily maintainable code; a value between 65 and 85 indicates average maintainability for the analyzed code; a value under 65 indicates hardly maintainable code. With the original and SEI formulas, the MI value can also be negative. With the Visual Studio formula, the thresholds for medium and high maintainability are moved respectively to 10 and 20.

The Maintainability Index is the highest-level metric considered in this study, as it includes an intermediate computation of one of the Halstead suite metrics.

## Objects

For the study, it was necessary to gather a set of simple code artifacts to analyze the Rust source code properties and compare them with other programming languages.

To that end, a set of nine simple algorithms was collected. In the set, each algorithm was implemented in five different languages: C, C++, JavaSript, Python, Rust, and TypeScript. All implementations of the code artifacts have been taken from the Energy-Languages repository (https://github.com/greensoftwarelab/Energy-Languages). The rationale behind the repository selection is its continuous and active maintenance and the fact that these code artifacts are adopted by various other projects for tests and benchmarking

**Table 7 Selected source code artifacts for the study.**

| Name | Description |
| --- | --- |
| binarytrees | Allocate and deallocate binary trees |
| fannkuchredux | Indexed-access to tiny integer-sequence |
| fasta | Generate and write random DNA sequences |
| knucleotide | Hashtable update and k-nucleotide strings |
| mandelbrot | Generate Mandelbrot set portable bitmap file |
| nbody | Double-precision N-body simulation |
| regexredux | Match DNA 8-mers and substitute magic patterns |
| revcomp | Read DNA sequences—write their reverse-complement |
| spectralnorm | Eigenvalue using the power method |

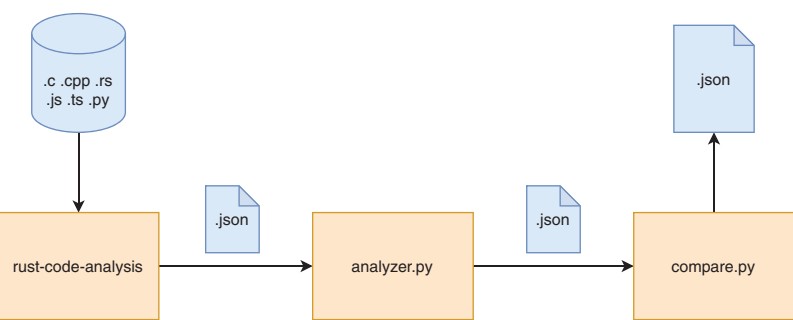

**Figure 1  Representation of the data flow of the framework.**

purposes, especially for evaluations of the execution speed of code written in a given programming language after compilation.

The number of different programming languages for the comparison was restricted to five because those languages (additional details are provided in the next section) were the common ones for the Energy-Languages repository and the set of languages that are correctly parsed by the tooling employed in the experiment conduction.

Table 7 lists the code artifacts used (sorted out alphabetically) and provides a brief description of each of them.

## Instruments

This section provides details about the framework that was developed to compare the selected metrics and the existing tools that were employed for code parsing and metric computation.

A graphic overview of the framework is provided in Fig. 1. The diagram only represents the logical flow of the data in the framework since the actual flow of operations is reversed, being the *compare.py* script the entry point of the whole computation. The rust-code-analysis tool is used to compute static metrics and save them in the JSON format.

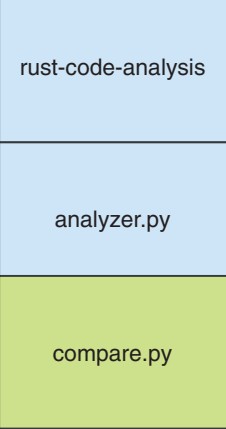

**Figure 2** Representation of the process stack of the framework.

The *analyzer.py* script receives as input the results in JSON format provided by the rust-code-analysis tool and formats them in a common notation that is more focused on academic facets of the computed metrics rather than the production ones used by the rust-code-analysis default formatting. The *compare.py* has been developed to call the *analyzer.py* script and to use its results to perform pair-by-pair comparisons between the JSON files obtained for source files written in different programming languages. These comparison files allow us to immediately assess the differences in the metrics computed by the different programming languages on the same software artifacts. The stack of commands that are called in the described evaluation framework is shown in Fig. 2.

The evaluation framework has been made available as an open-source repository on GitHub (https://github.com/SoftengPoliTo/SoftwareMetrics).

### The Rust code analysis tool

All considered metrics have been computed by adopting and extending a tool developed in the Rust language, and able to compute metrics for many different ones, called *rust-code-analysis*. We have forked version 0.0.18 of the tool to fix a few minor defects in metric computation and to uniform the presentation of the results, and we have made it available on a GitHub repository (https://github.com/SoftengPoliTo/rust-code-analysis).

We have decided to adopt and personally extend a project written in Rust because of the advantages guaranteed by this language, such as memory and thread safety, memory efficiency, good performance, and easy integration with other programming languages.

*rust-code-analysis* builds, through the use of an open-source library called *tree-sitter* (https://tree-sitter.github.io/), builds an Abstract Syntax Tree (AST) to represent the syntactic structure of a source file. An AST differs from a Concrete Syntax Tree because it does not include information about the source code less important details, like punctuation and parentheses. On top of the generated AST, *rust-code-analysis* performs a division of the source code in *spaces*, that is, any structure that can incorporate a function. It contains a series of fields such as the name of the structure, the relative line start,

```
 1   {
 2       "name": "Assets/Rust/binarytrees.rs",
 3       "start_line": 1,
 4       "end_line": 75,
 5       "kind": "unit",
 6       "metrics": {
 7           "nargs": {
 8               "sum": 14.0,
 9               "average": 2.0
10           },
11           "nexits": {
12               "sum": 3.0,
13               "average": 0.42857142857142855
14           },
15           "cognitive": {
16               "sum": 5.0,
17               "average": 0.7142857142857143
18           },
19           "cyclomatic": {
20               "sum": 12.0,
21               "average": 1.5
22           },
23           "halstead": {
24               "n1": 22.0,
25               "N1": 193.0,
26               "n2": 43.0,
27               "N2": 140.0,
28               "length": 333.0,
29               "estimated_program_length": 331.4368800622107,
30               "purity_ratio": 0.9953059461327649,
31               "vocabulary": 65.0,
32               "volume": 2005.4484817384753,
33               "difficulty": 35.81395348837209,
34               "level": 0.02792207792207792,
35               "effort": 71823.03864830818,
36               "time": 3990.168813794899,
37               "bugs": 0.5759541722145377
38           },
39           "loc": {
40               "sloc": 75.0,
41               "ploc": 56.0,
42               "lloc": 31.0,
43               "cloc": 7.0,
44               "blank": 12.0
45           },
46           "nom": {
47               "functions": 4.0,
48               "closures": 3.0,
49               "total": 7.0
50           },
51           "mi": {
52               "mi_original": 58.75785297946959,
53               "mi_sei": 33.08134287773029,
54               "mi_visual_studio": 34.36131753185356
55           }
56       }
57   }
```

**Listing 1** Sample output of the *rust-code-analysis* tool for the Rust version of the binarytrees algorithm.                               

line end, kind, and a *metric* object, which is composed of the values of the available metrics computed by *rust-code-analysis* on the functions contained in that space. All metrics computed at the function level are then merged at the parent space level, and this procedure continues until the space representing the entire source file is reached.

The tool is provided with parser modules that are able to construct the AST (and then to compute the metrics) for a set of languages: C, C++, C#, Go, JavaScript, Python, Rust, Typescript. The programming languages currently implemented in rust-code-analysis have been chosen because they are the ones that compose the Mozilla-central repository, which contains the code of the Firefox browser. The metrics can be computed for each language of this repository with the exception of Java, which does not have an implementation yet, and HTML and CSS, which are excluded because they are formatting languages.

*rust-code-analysis* can receive either single files or entire directories, detect whether they contain any code written in one of its supported languages, and output the resultant static metrics in various formats: textual, JSON, YAML, toml, cbor (*Ardito et al., 2020a*).

Concerning the original implementation of the rust-code-analysis tool, the project was forked and modified by adding metrics computations (e.g., the COGNITIVE metric). Also, the possible output format provided by the tool was changed.

Listing 1, reports an excerpt of the JSON file produced as output by rust-code-analysis.

### Analyzer

A Python script named *analyzer.py* was developed to analyze the metrics computed from rust-code-analysis. This script can launch different software libraries to compute metrics and adapt their results to a common format.

In this experiment, the *analyzer.py* script was used only with the Rust-code-analysis tool, but in a future extension of this study—or other empirical assessments—the script can be used to launch different tools simultaneously on the same source code.

The *analyzer.py* script performs the following operations:

- The arguments are parsed to verify their correctness. For instance, *analyzer.py* receives as arguments the list of tools to be executed, the path of the source code to analyze, and the path to the directory where to save the results;
- The selected metric computation tool(s) is (are) launched, to start the computation of the software metrics on the source files passed as arguments to the analyzer script;
- The output of the execution of the tool(s) is converted in JSON and formatted in order to have a common standard to compare the measured software metrics;
- The newly formatted JSON files are saved in the directory previously passed as an argument to *analyzer.py*.

The output produced by rust-code-analysis through *analyzer.py* was modified for the following reasons:

- The names of the metrics computed by the tool are not coherent with the ones selected from the scientific literature about software static quality metrics;
- The types of data representing the metrics are floating-point values instead of integers since rust-code-analysis aims at being as versatile as possible;
- The missing aggregation of each source file metrics contained in a directory within a single JSON-object, which is composed of global metrics and the respective metrics for each file. This additional aggregate data allows obtaining a more general prospect on the quality of a project written in a determined programming language.

Listing 2 reports an excerpt of the JSON file produced as output by the Analyzer script. As further documentation of the procedure, the full JSON files generated in the evaluation can be found in the Results folder of the project (https://github.com/SoftengPoliTo/SoftwareMetrics/tree/master/Results).

### Comparison

A second Python script, *Compare.py*, was finally developed to perform the comparisons over the JSON result files generated by the *Analyzer.py* script. The Compare.py script executes the comparisons between different language configurations, given an analyzed source code artifact and a metric.

The script receives a *Configuration* as a parameter, a pair of versions of the same code, written in two different programming languages.

```
 1  {
 2      "SLOC": 75,
 3      "PLOC": 56,
 4      "LLOC": 31,
 5      "CLOC": 7,
 6      "BLANK": 12,
 7      "CC_SUM": 12,
 8      "CC_AVG": 1.5,
 9      "COGNITIVE_SUM": 5,
10      "COGNITIVE_AVG": 0.7142857142857143,
11      "NARGS_SUM": 14,
12      "NARGS_AVG": 2.0,
13      "NEXITS": 3,
14      "NEXITS_AVG": 0.42857142857142855,
15      "NOM": {
16          "functions": 4,
17          "closures": 3,
18          "total": 7
19      },
20      "HALSTEAD": {
21          "n1": 22,
22          "n2": 43,
23          "N1": 193,
24          "N2": 140,
25          "Vocabulary": 65,
26          "Length": 333,
27          "Volume": 2005.4484817384753,
28          "Difficulty": 35.81395348837209,
29          "Level": 0.02792207792207792,
30          "Effort": 71823.03864830818,
31          "Programming_time": 3990.168813794899,
32          "Bugs": 0.5759541722145377,
33          "Estimated_program_length": 331.4368800622107,
34          "Purity_ratio": 0.9953059461327649
35      },
36      "MI": {
37          "Original": 58.75785297946959,
38          "Sei": 33.08134287773029,
39          "Visual_Studio": 34.36131753185356
40      }
41  }
```

**Listing 2** **Sample output of the *analyzer.py* script for the Rust version of the binarytrees algorithm.**

The script performs the following operations for each received *Configuration*:

- Computes the metrics for the two files of a configuration by calling the analyzer.py script;
- Loads the two JSON files from the Results directory and compares them, producing a JSON file of differences;
- Deletes all local metrics (the ones computed by rust-code-analysis for each subspace) from the JSON file of differences;
- Saves the JSON file of differences, now containing only global file metrics, in a defined destination directory.

The JSON differences file is produced using a JavaScript program called JSON-diff (https://www.npmjs.com/package/json-diff).

Listing 3 reports an excerpt of the JSON file produced as output by the Comparison script. As further documentation of the procedure, the full JSON files generated in the evaluation can be found in the Compare folder of the project (https://github.com/SoftengPoliTo/SoftwareMetrics/tree/master/Compare).

## Data collection and Analysis procedure

To collect the data to analyze, the described instruments were applied on each of the selected software objects for all the languages studied (i.e., for a total of 45 software artifacts).

The collected data was formatted in a single .csv file containing all the measurements.

To analyze the results, comparative analyses of the average and median of each of the measured metrics were performed to provide a preliminary discussion.

A non-parametric Kruskal–Wallis test was later applied to identify statistically significant differences among the different sets of metrics for each language.

```
 1   {
 2       "SLOC": {
 3           "__old": 139,
 4           "__new": 75
 5       },
 6       "PLOC": {
 7           "__old": 98,
 8           "__new": 56
 9       },
10       "LLOC": {
11           "__old": 25,
12           "__new": 31
13       },
14       "CLOC": {
15           "__old": 15,
16           "__new": 7
17       },
18       "BLANK": {
19           "__old": 26,
20           "__new": 12
21       },
22       "CC_SUM": {
23           "__old": 19,
24           "__new": 12
25       },
26       "CC_AVG": {
27           "__old": 1.4615384615384615,
28           "__new": 1.5
29       },
30       "COGNITIVE_SUM": {
31           "__old": 8,
32           "__new": 5
33       },
34       "COGNITIVE_AVG": {
35           "__old": 0.8888888888888888,
36           "__new": 0.7142857142857143
37       },
38       "NARGS_SUM": {
39           "__old": 2,
40           "__new": 14
41       },
42       "NARGS_AVG": {
43           "__old": 0.2222222222222222,
44           "__new": 2
45       },
46       "NEXITS": {
47           "__old": 5,
48           "__new": 3
49       },
50       "NEXITS_AVG": {
51           "__old": 0.5555555555555556,
52           "__new": 0.42857142857142855
53       },
54       "NOM": {
55           "functions": {
56               "__old": 9,
57               "__new": 4
58           },
59           "closures": {
60               "__old": 0,
61               "__new": 3
62           },
63           "total": {
64               "__old": 9,
65               "__new": 7
66           }
67       },
68       "HALSTEAD": {
69           "n1": {
70               "__old": 28,
71               "__new": 22
72           },
73           "n2": {
74               "__old": 56,
75               "__new": 43
76           },
77           "N1": {
78               "__old": 251,
79               "__new": 193
80           },
81           "N2": {
82               "__old": 173,
83               "__new": 140
84           },
85           "Vocabulary": {
86               "__old": 84,
87               "__new": 65
88           },
89           "Length": {
90               "__old": 424,
91               "__new": 333
92           },
93           "Volume": {
94               "__old": 2710.3425872581947,
95               "__new": 2005.4484817384753
96           },
97           "Difficulty": {
98               "__old": 43.25,
99               "__new": 35.81395348837209
100          },
101          "Level": {
102              "__old": 0.023121387283236993,
103              "__new": 0.02792207792207792
104          },
105          "Effort": {
106              "__old": 117222.31689891692,
107              "__new": 71823.03864830818
108          },
109          "Programming_time": {
110              "__old": 6512.3509388287175,
111              "__new": 3990.168813794899
112          },
113          "Bugs": {
114              "__old": 0.7983970910222301,
115              "__new": 0.5759541722145377
116          },
117          "Estimated_program_length": {
118              "__old": 459.81781345283866,
119              "__new": 331.4368800622107
120          },
121          "Purity_ratio": {
122              "__old": 1.0844759751246196,
123              "__new": 0.9953059461327649
124          }
125      },
126      "MI": {
127          "Original": {
128              "__old": 45.586404609681736,
129              "__new": 58.75785297946959
130          },
131          "Sei": {
132              "__old": 16.3624350913677,
133              "__new": 33.08134287773029
134          },
135          "Visual_Studio": {
136              "__old": 26.658716146012715,
137              "__new": 34.36131753185356
138          }
139      }
140  }
```

**Listing 3 Sample output of the *compare.py* script for the C++/Rust comparisons of the binarytrees algorithm.** The *_old* label identifies C++ metric values, while new the Rust ones.

For significantly different distributions, post-hoc comparisons with Wilcoxon signed rank-sum test, with Benjamini-Hochberg correction (*Ferreira & Zwinderman, 2006*), were applied to analyze the difference between the metrics measured for Rust and the other five languages in the set.

Descriptive and statistical analyses and graph generation were performed in R. The data and scripts have been made available in an online repository (https://github.com/SoftengPoliTo/rust-analysis).

## Threats to validity

### Threats to internal validity

The study results may be influenced by the specific selection of the tool with which the software metrics were computed, namely the *rust-code-analysis* tool. The values measured for the individual metrics (and, by consequence, the reasoning based upon them) can be heavily influenced by the exact formula used for the metric computation.

In the Halstead suite, the formulas depend on two coefficients defined explicitly in the literature for every software language, namely the denominators for the T and B metrics. Since no previous result in the literature has provided Halstead coefficients specific to Rust, the C coefficients were used for the computation of Rust Halstead metrics. More specifically, 18 was used as the denominator of the T metric. This value, called Stoud number (S), is measured in moments, that is, the time required by the human brain to carry out the most elementary decision. In general, S is comprised between 5 and 20. In the original Halstead metrics suite for the C language, a value of 18 is used. This value was empirically defined after psychological studies of the mental effort required by coding. A total of 3,000 was selected as the denominator of the Number of delivered Bugs metric; this value, again, is the original value defined for the Halstead suite and represents the number of mental discriminations required to produce an error in any language. The 3,000 value was originally computed for the English language and then mutated for programming languages (*Ottenstein, Schneider & Halstead, 1976*). The choice of the Halstead parameters may significantly influence the values obtained for the T and B metrics. The definition of the specific parameters for a new programming language, however, implies the need for a thorough empirical evaluation of such parameters. Future extensions of this work may include studies to infer the optimal Halstead parameters for Rust source code.

Finally, two metrics, NARGS and NEXITS, were adopted for the evaluation of readability and organization of code. Albeit extensively used in production (they are used in the Mozilla-central open-source codebase), these metrics still miss empirical validation on large repositories, and hence their capacity of predicting code readability and complexity cannot be ensured.

### Threats to external validity

The results presented in this research have been measured on a limited number of source artifacts (namely, nine different code artifacts per programming language). Therefore, we acknowledge that the results cannot be generalized to all software written with one of

the analyzed programming languages. Another bias can be introduced in the results by the characteristics of the considered code artifacts. All considered source files were small programs collected from a single software repository. The said software repository itself was implemented for a specific purpose, namely the evaluation of the performance of different programming languages at runtime. Therefore, it is still unsure whether our measurements can scale up to bigger software repositories and real-world applications written in the evaluated languages. As well, the results of the present manuscript may inherit possible biases that the authors of the code had in writing the source artifacts employed for our evaluation. Future extensions of the current work should include the computation of the selected metrics on more extensive and more diverse sets of software artifacts to increase the generalizability of the present results.

### Threats to conclusion validity

The conclusions detailed in this work are only based on the analysis of quantitative metrics and do not consider other possible characteristics of the analyzed source artifacts (e.g., the developers' coding style who produced the code). Like the generalizability of the results, this bias can be reduced in future extensions of the study using a broader and more heterogeneous set of source artifacts (*Sjøberg, Anda & Mockus, 2012*).

In this work, we make assumptions on verbosity, complexity, understandability, and maintainability of source code based on quantitative static metrics. It is not ensured that our assumptions are reflected by maintenance and code understanding effort in real-world development scenarios. It is worth mentioning that there is no unanimous opinion about the ability of more complex metrics (like MI) to capture the maintainability of software programs more than simpler metrics like lines of code and Cyclomatic Complexity.

Researcher bias is a final theoretical threat to the validity of this study since it involved a comparison in terms of different metrics of different programming languages. However, the authors have no reason to favor any particular approach, neither inclined to demonstrate any specific result.

## RESULTS AND DISCUSSION

This section reports the results gathered by applying the methodology described in the previous section, subdivided according to the research question they answer.

### RQ1: code verbosity

The boxplots in Fig. 3 and Table 8 report the measures for the metrics adopted to answer RQ1.

The mean and median values of the Source Lines of Code (SLOC) metric (i.e., total lines of code in the source files) are largely higher for the C, C++, and Rust language: the highest mean SLOC was for C (209 average LOCs per source file), followed by C++ (186) and Rust (144). The mean values are way smaller for Python, TypeScript, and JavaScript (respectively, 98, 107, and 95).

A similar trend is assumed by the Physical Lines of Code (PLOC) metric, i.e., the total number of instructions and comment lines in the source files. In the examined set,

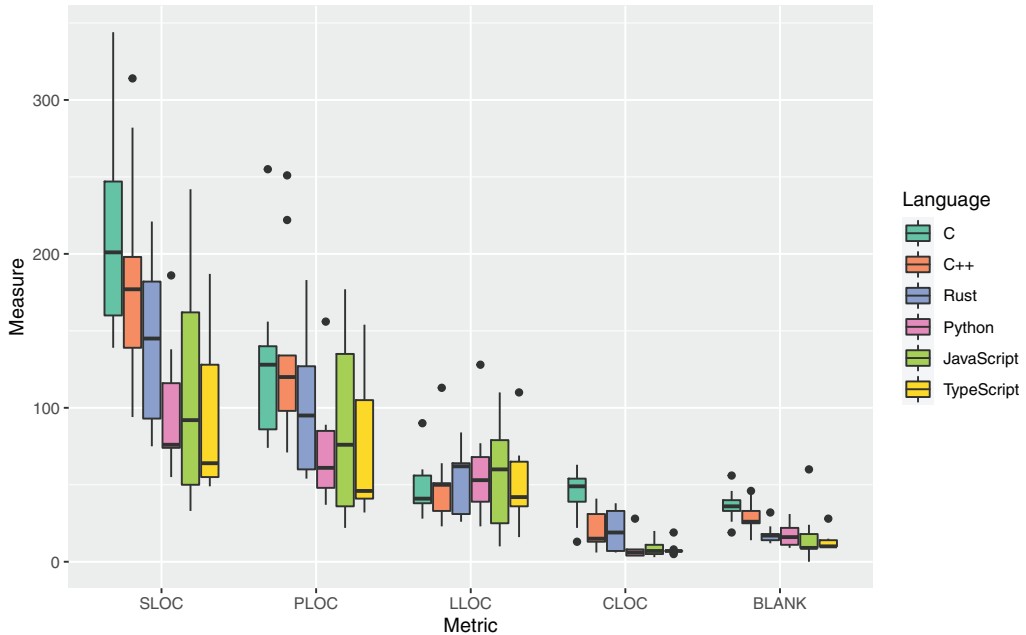

**Figure 3 Distribution of the metrics about lines of code for all the considered programming languages.**

**Table 8 Mean (Median) values of the metrics about lines of code for all the considered programming languages.**

| Language | SLOC | PLOC | LLOC | CLOC | BLANK |
|---|---|---|---|---|---|
| C | 209 (201) | 129 (128) | 48 (41) | 43 (49) | 37 (36) |
| C++ | 186 (177) | 137 (120) | 51 (50) | 20 (15) | 28 (26) |
| Rust | 144 (145) | 105 (95) | 52 (62) | 21 (19) | 18 (17) |
| Python | 99 (76) | 73 (61) | 59 (53) | 8 (6) | 18 (16) |
| JavaScript | 107 (92) | 83 (76) | 58 (60) | 9 (7) | 16 (9) |
| TypeScript | 95 (64) | 74 (46) | 51 (42) | 8 (7) | 13 (10) |

74 average PLOCs per file were measured for the Rust language. The highest and smallest values were again measured respectively for C and TypeScript, with 129 and 74 average PLOCs per file. The values measured for the CLOC and BLANK metrics showed that a higher number of empty lines of code and comments were measured for C than for all other languages. In the CLOC metric, the Rust language exhibited the second-highest mean of all languages, suggesting a higher predisposition of Rust developers at providing documentation in the developed source code.

A slightly different trend is assumed by the Logical Lines of Code (LLOC) metric (i.e., the number of instructions or statements in a file). In this case, the mean number of statements for Rust code is higher than the ones measured for C, C++ and TypeScript, while the SLOC and PLOC metrics are lower. The Rust scripts also had the highest median LLOC. This result may be influenced with the different number of types of statements

**Table 9 Null hypotheses and *p*-values for RQ1 metrics obtained by applying Kruskal–Wallis chi-squared test (Signific. codes: 0 "***" 0.001 "**" 0.01 "*" 0.05 "." 0.1 "–" 1).**

| Name | Description | *p*-value | Decision | Significance |
|------|-------------|-----------|----------|--------------|
| $H0_{\text{SLOC}}$ | No significant difference in SLOC for the sw artifacts | 0.001706 | Reject | ** |
| $H0_{\text{PLOC}}$ | No significant difference in PLOC for the artifacts | 0.03617 | Reject | * |
| $H0_{\text{LLOC}}$ | No significant difference in LLOC for the artifacts | 0.9495 | Not Reject | – |
| $H0_{\text{CLOC}}$ | No significant difference in CLOC for the artifacts | 7.07e−05 | Reject | *** |
| $H0_{\text{BLANK}}$ | No significant difference in BLANK for the artifacts | 0.0001281 | Reject | *** |

that are offered by the language. For instance, the Rust language provides 19 types of statements while C offers just 14 types (e.g., the Rust statements *If let* and *While let* are not present in C). The higher amount of logical statements may indeed hint at a higher decomposition of the instructions of the source code into more statements, that is, more specialized statements covering less operations.

Albeit many higher-level measures and metrics have been derived in the latest years by related literature to evaluate the understandability and maintainability of software, the analysis of code verbosity can be considered a primary proxy for these evaluations. In fact, several studies have linked the intrinsic verbosity of a language to lower readability of the software code, which translates to higher effort when the code has to be maintained. For instance, *Flauzino et al. (2018)* state that verbosity can cause higher mental energy in coders working on implementing an algorithm and can be correlated to many smells in software code. *Toomim, Begel & Graham (2004)* highlight that redundancy and verbosity can obscure meaningful information in the code, thereby making it difficult to understand.

The metrics for RQ1 where mostly evenly distributed among different source code artifacts. Two outliers were identified for the PLOC metric in C and C++ (namely, *fasta.c* and *fasta.cpp*), mostly due to the fact that they have the highest SLOC value, so the results are coherent. More marked outliers were found for the BLANK metric, but such measure is strongly influenced by the developer's coding style and the used code formatters; thereby, no valuable insight can be found by analyzing the individual code artifacts.

Table 9 reports the results of applying the Kruskal–Wallis non-parametric test on the set of measures for RQ1. The difference for SLOC, PLOC, CLOC, and BLANK were statistically significant (with strong significance for the last two metrics). Post-hoc statistical tests focused on the comparison between Rust, and the other languages (Table 10) led to the evidence that Rust had a significantly lower CLOC than C and a significantly lower BLANK than C and TypeScript.

**Answer to RQ1**: The examined source files written in Rust exhibited an average verbosity (144 mean SLOCs per file and 74 mean PLOCs per file). Such values are lower than C and C++ and higher than the other considered object-oriented languages. Rust exhibited the third-highest average (and highest median) LLOC among all considered

**Table 10** *p*-Values for post-hoc Wilcoxon signed rank test for RQ1 metrics between Rust and the other languages (significant *p*-values in bold).

| Metric | C | C++ | JavaScript | Python | TypeScript |
|---|---|---|---|---|---|
| SLOC | 0.0519 | 0.3309 | 0.2505 | 0.0420 | 0.0519 |
| PLOC | 0.3770 | 0.3081 | 0.3607 | 0.2790 | 0.2790 |
| CLOC | **0.0399** | 0.8242 | 0.0620 | 0.0620 | 0.097 |
| BLANK | **0.0053** | 0.0618 | 0.1944 | 0.7234 | **0.0467** |

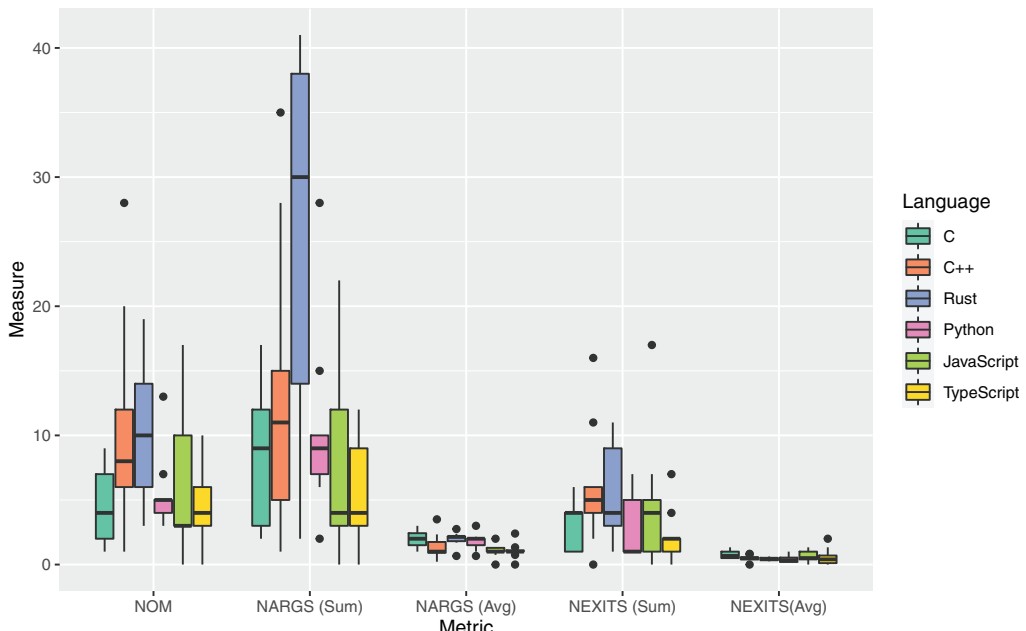

**Figure 4** **Distribution of the metrics about organization of code for all the considered programming languages.**

languages. Significantly lower values were measured for CLOC against C and BLANK against C and TypeScript.

## RQ2: code organization

The boxplots in Fig. 4 and Table 11 report the measures for the metrics adopted to answer RQ2. For each source file, two different measures were collected for the Number of Arguments (NARGS) metric: the sum at file level of all the methods arguments and the average at file level of the number of arguments per method (i.e., NARGS/NOM).

The Rust language had the highest median value for the Number of Methods (NOM) metric, with a median of 10 methods per source file. The average NOM value was only lower than the one measured for C++ sources. However, this value was strongly influenced by the presence of one outlier in the set of analyzed sources (namely, the C++ implementation of *fasta* having a NOM equal to 20). While the NOM values were similar for C++ and Rust, all other languages exhibited much lower distributions, with the lowest

**Table 11 Mean (Median) values of the metrics about code organization for all the considered programming languages.**

| Language | NOM | NARGS (Sum) | NARGS (Avg) | NEXITS (Sum) | NEXITS (Avg) |
|---|---|---|---|---|---|
| C | 4.4 (4) | 8.6 (9) | 2.0 (2) | 3.1 (4) | 0.75 (0.67) |
| C++ | 10.6 (8) | 13.4 (11) | 1.4 (1) | 6.0 (5) | 0.48 (0.5) |
| Rust | 10.3 (10) | 25.1 (30) | 2.0 (2) | 5.7 (4) | 0.44 (0.43) |
| Python | 5.7 (5) | 10.6 (9) | 1.8 (2) | 2.8 (1) | 0.45 (0.33) |
| JavaScript | 5.9 (3) | 7.4 (4) | 1.1 (1) | 4.6 (4) | 0.63 (0.5) |
| TypeScript | 4.7 (4) | 5.7 (4) | 1.1 (1) | 2.1 (2) | 0.58 (0.4) |

median value for JavaScript (3). This high number of Rust methods can be seen as evidence of higher modularity than the other languages considered.

Regarding the number of arguments, it can be noticed that the Rust language exhibited the highest average and median cumulative number of arguments (Sum of Arguments) of all languages. The already discussed high NOM value influences this result. The highest NOM (and, by consequence, of the total cumulative number of arguments) can be caused by the missing possibility of having default values in the Rust language. This characteristic may lead to multiple variations of the same method to take into account changes in the parameter, thereby leading to a higher NARGS. The lowest average measures for NOM and NARGS_Sum metrics were obtained for the C language. This result can be justified by the lower modularity of the C language. By examining the C source files, it could be verified that the code presented fewer functions and more frequent usage of nested loops, while the Rust sources were using more often data structures and ad-hoc methods. In general, the results gathered for these metrics suggest a more structured Rust code organization with respect to the C language.

The NOM metric has an influence on the verbosity of the code, and therefore it can be considered as a proxy of the readability and maintainability of the code.

Regarding the Number of Exits (NEXITS) metric, the values were close for most languages, except Python and TypeScript, which respectively contain more methods without exit points and fewer functions. The obtained NEXITS value for Rust shows many exit points distributed among many functions, as demonstrated by the NOM value, making the code much more comfortable to follow.

An analysis of the outliers of the distributions of the measurements for RQ2 was performed. For C++, the highest value of NOM was exhibited by the *revcomp.cpp* source artifact. This high value was caused by the extensive use of classes methods to handle chunks of DNA sequences. *knucleotide.py* and *spectralnorm.py* had a higher number of functions than the other considered source artifacts. *fasta.cpp* uses lots of mall functions with many arguments, resulting in an outlier value for the $\text{NARGS}_{\text{SUM}}$ metric. *pidigits.py* had 0 values for NOM and NARGS, since it used zero functions. Regarding NEXITS, very high values were measured for *fasta.cpp* and *revcomp.cpp*, which had many functions with return statements. Lower values were measured for *regexredup.cpp*, which has a single main function without any return, and *pidigits.cpp*, which has a single return. A final

**Table 12 Null hypotheses and *p*-values for RQ2 metrics obtained by applying Kruskal–Wallis chi-squared test (Signific. codes: 0 "\*\*\*" 0.001 "\*\*" 0.01 "\*" 0.05 "." 0.1 "–" 1).**

| Name | Description | *p*-value | Decision | Significance |
|---|---|---|---|---|
| $H0_{NOM}$ | No significant difference in NOM for the artifacts | 0.04372 | Reject | * |
| $H0_{NARGSSUM}$ | No significant difference in $NARGS_{SUM}$ for the artifacts | 0.02357 | Reject | * |
| $H0_{NARGSAVG}$ | No significant difference in $NARGS_{AVG}$ for the artifacts | 0.008224 | Reject | ** |
| $H0_{NEXITSSUM}$ | No significant difference in $NEXITS_{SUM}$ for the artifacts | 0.142 | Not Reject | – |
| $H0_{NEXITSAVG}$ | No significant difference in $NEXITS_{AVG}$ for the artifacts | 0.2485 | Not Reject | – |

**Table 13 *p*-Values for post-hoc Wilcoxon signed rank test for RQ2 metrics between Rust and the other languages (significant *p*-values in bold).**

| Metric | C | C++ | JavaScript | Python | TypeScript |
|---|---|---|---|---|---|
| NOM | 0.0534 | 0.7560 | 0.1037 | 0.0546 | 0.0533 |
| $NARGS_{SUM}$ | **0.0239** | 0.0633 | **0.0199** | **0.0318** | **0.0177** |
| $NARGS_{AVG}$ | 0.5658 | 0.1862 | **0.0451** | 0.4392 | 0.0662 |

outlier was the NEXITS value for *fasta.js*, which features a very high number of function with return statements.

Table 12 reports the results of the application of the Kruskal–Wallis non-parametric test on the set of measures for RQ2. The difference for NOM, $NARGS_{SUM}$ and $NARGS_{AVG}$ was statistically significant, while no significance was measured fo the metrics related to the NEXITS. Post-hoc statistical tests focused on the comparison between Rust, and the other languages (Table 13) highlighted that Rust had a significantly higher $NARGS_{SUM}$ than C, JavaScript, Python, and TypeScript, and a significantly higher $NARGS_{AVG}$ than JavaScript.

**Answer to RQ2**: The examined source files written in Rust exhibited the most structured organization of the considered set of languages (with a mean 10.3 NOM per file, with a mean of 2 arguments for each method). The Rust language had a significantly higher number of arguments than C, JavaScript, Python, and TypeScript.

## RQ3: code complexity

The boxplots in Fig. 5 and Table 14 report the measures for the metrics adopted to answer RQ3. For the Computational Complexity, two metrics were computed: the sum of the Cyclomatic Complexity (CC) of all *spaces* in a source file ($CC_{Sum}$), and the averaged value of CC over the number of spaces in a file ($CC_{Avg}$). A space is defined in *rust-code-analysis* as any structure that incorporates a function. For what concerns the COGNITIVE complexity, two metrics were computed: the sum of the COGNITIVE complexity associated to each function and closure present in a source file, ($COGNITIVE_{Sum}$), and the average value of COGNITIVE complexity, ($COGNITIVE_{Avg}$), always computed over the number of functions and closures. Table 14 reports the mean and median values over the set of different source files selected for each language, of the sum and average metrics computed at the file level.

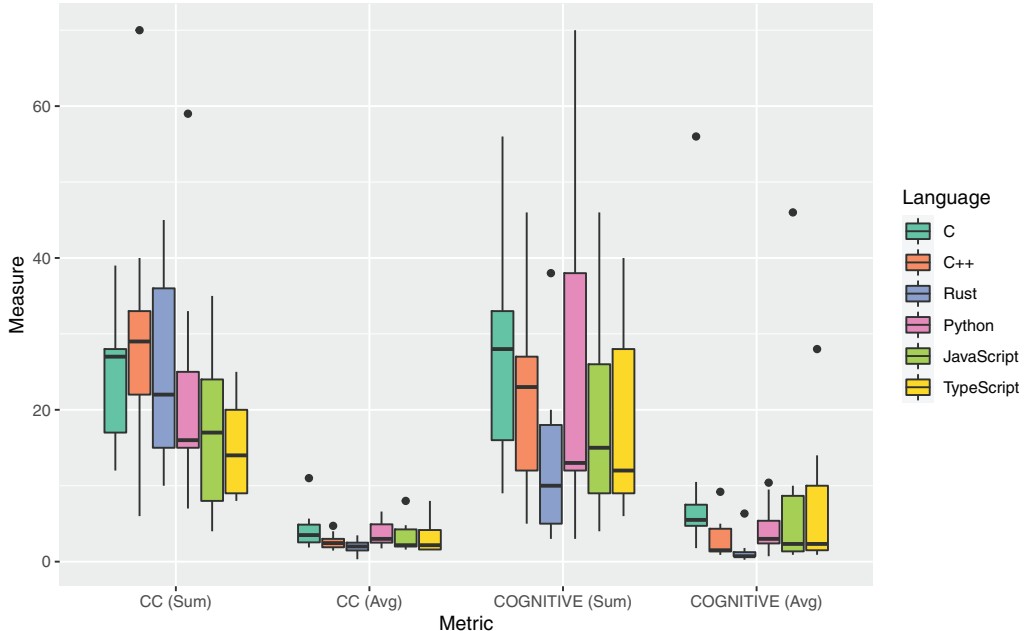

**Figure 5 Distribution of complexity metrics for all the considered programming languages.**

**Table 14 Mean (Median) values of the complexity metrics for all the considered programming languages.**

| Language | $CC_{Sum}$ | $CC_{Avg}$ | $COGNITIVE_{Sum}$ | $COGNITIVE_{Avg}$ |
|---|---|---|---|---|
| C | 27.3 (28) | 4.3 (3.5) | 24.3 (21.0) | 11.2 (5.5) |
| C++ | 31.1 (29) | 2.7 (2.4) | 22.4 (23.0) | 3.2 (1.5) |
| Rust | 25.3 (22) | 2.0 (2.0) | 13.1 (10.0) | 1.5 (0.7) |
| Python | 23.0 (16) | 3.6 (3.0) | 25.4 (13.0) | 4.4 (3.0) |
| JavaScript | 17.6 (17) | 3.4 (2.2) | 19.9 (15.0) | 8.5 (2.3) |
| TypeScript | 15.2 (14) | 3.4 (2.2) | 17.0 (12.0) | 7.2 (2.3) |

As commonly accepted in the literature and practice, a low cyclomatic complexity generally indicates a method that is easy to understand, test, and maintain. The reported measures showed that the Rust language had a lower median $CC_{Sum}$ (22) than C and C++ and the second-highest average value (25.3). The lowest average and median $CC_{Sum}$ was measured for the TypeScript language. By considering the average of the Cyclomatic Complexity, $CC_{Avg}$, at the function level, the highest average and mean values are instead obtained for the Rust language. It is worth mentioning that the average CC values for all the languages were rather low, hinting at an inherent simplicity of the software functionality under examination. So an analysis based on different codebases may result in more pronounced differences between the programming languages.

COGNITIVE complexity is a software metric that assesses the complexity of code starting from human judgment and is a measure for source code comprehension by the developers and maintainers (*Barón, Wyrich & Wagner, 2020*). Moreover, empirical results

have also proved the correlation between COGNITIVE complexity and defects (*Alqadi & Maletic, 2020*). For both the average COGNITIVE complexity and the sum of COGNITIVE complexity at the file level, Rust provided the lowest mean and median values. Specifically, Rust guaranteed a COGNITIVE complexity of 0.7 per method, which is less than half the second-lowest value for C++ (1.5). The highest average COGNITIVE complexity per class was measured for C code (5.5). This very low value of the COGNITIVE complexity per method for Rust is related to the highest number of methods for Rust code (described in the analysis of RQ2 results). By considering the sum of the COGNITIVE complexity metric at the file level, Rust had a mean COGNITIVE$_{Sum}$ of 13.1 over the 9 analyzed source files. The highest mean value for this metric was measured for Python (25.4), and the highest median for C++ (23). Such lower values for the Rust language can suggest a more accessible, less costly, and less prone to bug injection maintenance for source code written in Rust. This lowest value for the COGNITIVE metric counters some measurements (e.g., for the LLOC and NOM metrics) by hinting that the higher verbosity of the Rust language has not a visible influence on the readability and comprehensibility of the Rust code.

The boxplots in Fig. 6 and Table 15 report the distributions, mean, and median of the Halstead metrics computed for the six different programming languages.

The Halstead Difficulty (D) is an estimation of the difficulty of writing a program that is statically analyzed. The Difficulty is the inverse of the program level metric. Hence, as the volume of the implementation of code increases, the difficulty increases as well. The usage of redundancy hence influences the Difficulty. It is correlated to the number of operators and operands used in the code implementation. The results suggest that the Rust programming language has an average Difficulty (median of 45.9) on the set of considered languages. The most difficult code to interpret, according to Halstead metrics, was C (median of 55.9), while the easiest to interpret was Python (median of 30.0). A similar hierarchy between the different languages is obtained for the Halstead Effort (E), which estimates the mental activity needed to translate an algorithm into code written in a specific language. The Effort is linearly proportional to both Difficulty and Volume. The unit of measure of the metric is the number of elementary mental discriminations (*Halstead, 1977*).

The Halstead Length (L) metric is given by the total number of operator occurrences and the total number of operand occurrences. The Halstead Volume (V) metric is the information content of the program, linearly dependent on its vocabulary. Rust code had the third-highest mean and median Halstead Length (602.2 mean, 550.0 median) and Halstead Volume (4,032 mean, 3,610 median), again below those measured for C and C++. The results measured for all considered source files were in line with existing programming guidelines (Halstead Volume lower than 8,000). The reported results about Length and Volume were, to some extent, expectable since these metrics are largely correlated to the number of lines of code present in a source file (*Tashtoush, Al-Maolegi & Arkok, 2014*).

The Halstead Time metric (T) is computed as the Halstead Effort divided by 18. It estimates the time in seconds that it should take a programmer to implement the code. A mean and median T of 11,064 and 13,719 seconds were measured, respectively, for the

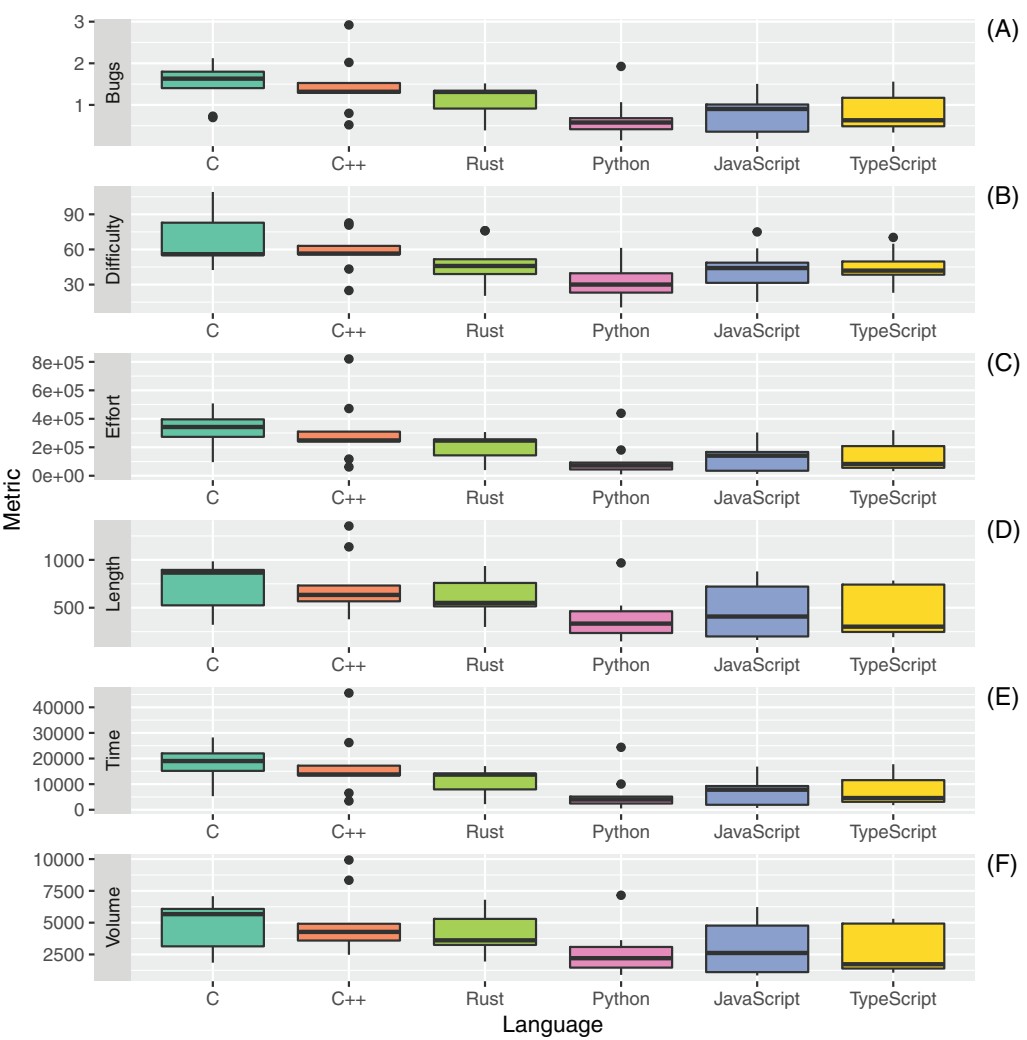

**Figure 6** Distribution of Halstead metrics ((A) Bugs; (B) Difficulty; (C) Effort; (D) Length; (E) Time and (F) Volume) for all the considered programming languages.

**Table 15** Mean (Median) values of Halstead metrics for all the considered programming languages.

| Language | Bugs | Difficulty | Effort | Length | Programming time | Volume |
|---|---|---|---|---|---|---|
| C | 1.52 (1.6) | 66.7 (55.9) | 322,313 (342,335) | 726.0 (867.0) | 17,906 (19,018) | 4,819 (5,669) |
| C++ | 1.46 (1.3) | 57.8 (56.4) | 311,415 (248,153) | 728.1 (634.0) | 17,300 (13,786) | 4,994 (4,274) |
| Rust | 1.1 (1.3) | 48.6 (45.9) | 199,152 (246,959) | 602.2 (550.0) | 11,064 (13,719) | 4,032 (3610) |
| Python | 0.7 (0.6) | 33.7 (30.0) | 111,103 (72,110) | 393.8 (334.0) | 6,172 (4,006) | 2,680 (2204) |
| JavaScript | 0.8 (0.9) | 43.1 (44.1) | 139,590 (140,951) | 458.6 (408.0) | 7,755 (7,830) | 2,963 (2615) |
| TypeScript | 0.8 (0.6) | 45.2 (41.9) | 132,644 (82,369) | 435.7 (302.0) | 7,369 (4,576) | 2,734 (1730) |

**Table 16 Null hypotheses and *p*-values for RQ3 metrics obtained by applying Kruskal–Wallis chi-squared test (Signific. codes: 0 "\*\*\*" 0.001 "\*\*" 0.01 "\*" 0.05 "." 0.1 "–" 1).**

| Name | Description | *p*-value | Decision | Significance |
|------|-------------|-----------|----------|--------------|
| $H0_{\text{CC SUM}}$ | No significant difference in $CC_{\text{SUM}}$ for the artifacts | 0.113 | Not reject | – |
| $H0_{\text{CC AVG}}$ | No significant difference in $CC_{\text{AVG}}$ for the artifacts | 0.1309 | Not Reject | – |
| $H0_{\text{COGNITIVE SUM}}$ | No significant difference in $COGNITIVE_{\text{SUM}}$ for the artifacts | 0.4554 | Not Reject | – |
| $H0_{\text{COGNITIVE AVG}}$ | No significant difference in $COGNITIVE_{\text{AVG}}$ for the artifacts | 0.009287 | Reject | \*\* |
| $H0_{\text{Halstead Vocabulary}}$ | No significant difference in Halstead Vocabulary for the artifacts | 0.07718 | Not Reject | . |
| $H0_{\text{Halstead Difficulty}}$ | No significant difference in Halstead Difficulty for the artifacts | 0.01531 | Reject | \* |
| $H0_{\text{Halstead Prog.time}}$ | No significant difference in Halstead Prog. time for the artifacts | 0.005966 | Reject | \*\* |
| $H0_{\text{Halstead Effort}}$ | No significant difference in Halstead Effort for the artifacts | 0.005966 | Reject | \*\* |
| $H0_{\text{Halstead Volume}}$ | No significant difference in Halstead Volume for the artifacts | 0.03729 | Reject | \* |
| $H0_{\text{Halstead Bugs}}$ | No significant difference in Halstead Bugs for the artifacts | 0.005966 | Reject | \*\* |

Rust programming language. These values are significantly distant from those measured for Python and TypeScript (the lowest) and from those measured for C and C++ (the highest).

Finally, the Halstead Bugs Metric estimates the number of bugs that are likely to be found in the software program. It is given by a division of the Volume metric by 3,000. We estimated a mean value of 1.1 (median 1.3) bugs per file with the Rust programming language on the considered set of source artifacts.

An analysis of the outliers of the distributions of measurements regarding RQ3 was performed. A relevant outlier for the CC metric was *revcomp.cpp*, in which the usage of many nested loops and conditional statements inside class methods significantly increased the computed complexity. For the set of Python source files, *knucleoutide.py* had the highest CC due to the usage of nested code; the same effect occurred for *fannchuckredux.rs* which had the highest CC and COGNITIVE complexity for the Rust language. The JavaScript and TypeScript versions of *fannchuckredux* both presented a high usage of nested code, but the lower level of COGNITIVE complexity for the TypeScript version suggests a better-written source code artifact. The few outliers that were found for the Halstead metrics measurements were principally for C++ source artifacts and mostly related to the higher PLOC and number of operands of the C++ source codes.

Table 16 reports the results of the application of the Kruskal-Wallis non-parametric test on the set of measures for RQ3. No statistical significance was measured for the differences in the measurements of the two metrics related to CC. A statistically significant difference was measured for the averaged COGNITIVE complexity. Regarding the Halstead metrics, all differences were statistically significant with the exception of those for the *Difficulty* metric. Post-hoc statistical tests focused on the comparison between Rust and the other languages (Table 17) highlighted that Rust had a significantly lower average COGNITIVE complexity than all the other considered languages.

**Answer to RQ3**: The Rust software artifacts exhibited an average Cyclomatic Complexity (mean 2.0 per function) and a significantly lower COGNITIVE complexity

**Table 17 p-Values for post-hoc Wilcoxon signed rank test for RQ3 metrics between Rust and the other languages (significant p-values in bold).**

| Metric | C | C## | JavaScript | Python | TypeScript |
|---|---|---|---|---|---|
| COGNITIVE$_{AVG}$ | **0.0062** | **0.0244** | **0.0222** | **0.0240** | **0.0222** |
| HALSTEAD$_{Difficulty}$ | 0.2597 | 0.2621 | 0.5328 | 0.2621 | 0.6587 |
| HALSTEAD$_{Programming\ Time}$ | 0.1698 | 0.3767 | 0.3081 | 0.1930 | 0.3134 |
| HALSTEAD$_{Effort}$ | 0.1698 | 0.3767 | 0.3081 | 0.1930 | 0.3134 |
| HALSTEAD$_{Volume}$ | 0.5960 | 0.5328 | 0.2621 | 0.2330 | 0.2330 |
| HALSTEAD$_{Bugs}$ | 0.1698 | 0.3767 | 0.3081 | 0.1930 | 0.3134 |

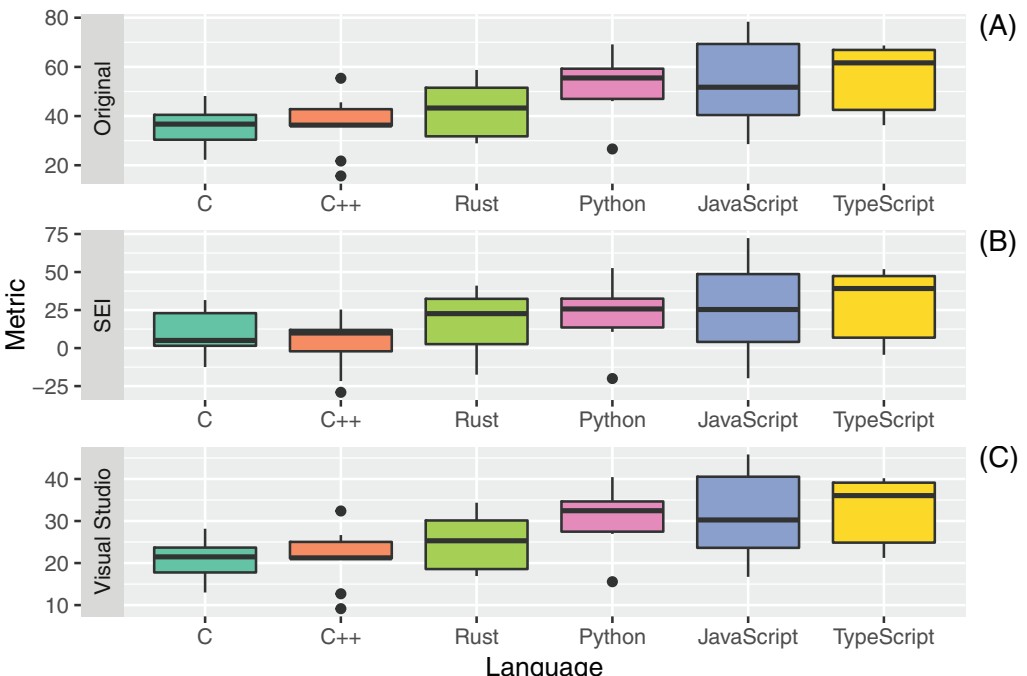

**Figure 7 Distribution of maintainability indexes ((A) Original; (B) SEI and (C) Visual Studio) for all the considered programming languages.**

(mean 1.5 per function) than all other languages. Rust was the third-highest performing language, after C and C++, for the Halstead metric values.

## RQ4: code maintainability

The boxplots in Fig. 7 and Table 18 report the distributions, mean, and median of the Maintainability Indexes computed for the six different programming languages.

The Maintainability Index is a composite metric aiming to give an estimate of software maintainability over time. The Metric has correlations with the Halstead Volume (V), the Cyclomatic Complexity (CC), and the number of lines of code of the source under examination.

**Table 18 Mean (Median) values of maintainability indexes for all the considered programming languages.**

| Language | Original | SEI | Visual studio |
|---|---|---|---|
| C | 35.9 (36.7) | 10.5 (5.0) | 21.0 (21.5) |
| C++ | 36.5 (36.3) | 3.6 (9.9) | 21.3 (21.2) |
| Rust | 43.0 (43.3) | 15.8 (22.6) | 25.1 (25.3) |
| Python | 52.5 (55.5) | 23.3 (25.7) | 30.7 (32.5) |
| JavaScript | 54.2 (51.7) | 27.7 (25.3) | 31.7 (30.3) |
| TypeScript | 55.9 (61.6) | 29.4 (39.2) | 32.7 (36.0) |

**Table 19 Null hypotheses and $p$-values for RQ4 metrics obtained by applying Kruskal–Wallis chi-squared test (Signific. codes: 0 "***" 0.001 "**" 0.01 "*" 0.05 "." 0.1 "–" 1).**

| Name | Description | $p$-value | Decision | Significance |
|---|---|---|---|---|
| $H0_{\text{MI Original}}$ | No significant difference in MI Original for the artifacts | 0.006002 | Reject | ** |
| $H0_{\text{MI SEI}}$ | No significant difference in MI SEI for the artifacts | 0.1334 | Not Reject | . |
| $H0_{\text{MI Visual Studio}}$ | No significant difference in MI Visual Studio for the artifacts | 0.006002 | Reject | ** |

The source files written in Rust had an average MI that placed the fourth among all considered programming languages, regardless of the specific formula used for the calculation of the MI. Minor differences in the placement of other languages occurred, for example, the median MI for C is higher than for C++ with the original formula for the Maintainability Index and lower with the SEI formula. Regardless of the formula used to compute MI, the highest maintainability was achieved by the TypeScript language, followed by Python and JavaScript. These results were expectable in light of the previous metrics measured, given the said strong dependency of the MI on the raw size of source code.

It is interesting to underline that, in accordance with the original guidelines for the MI computation, all the values measured for the software artifacts under study would suggest hard to maintain code, being the threshold for easily maintainable code set to 80. On the other hand, according to the documentation of the Visual Studio MI metric, all source artifacts under test can be considered as easy to maintain ($\text{MI}_{\text{VS}}$ 20).

Outliers in the distributions of MI values were mostly found for C++ sources and were likely related to higher values of SLOC, CC, and Halstead Volume, all leading to very low MI values.

Table 19 reports the results of the application of the Kruskal-Wallis non-parametric test on the set of measures for RQ4. The measured differences were statistically significant for the original MI metric and for the version employed by Visual Studio. Post-hoc statistical tests focused on the comparison between Rust, and the other languages (Table 20) highlighted that difference was statistically significant.

**Answer to RQ4**: Rust exhibited an average Maintainability Index, regardless of the specific formula used (median values of 43.3 for $\text{MI}_{\text{O}}$, 22.6 for $\text{MI}_{\text{SEI}}$, 25.3 for $\text{MI}_{\text{VS}}$). Highest Maintainability index were obtained for Python, JavaScript and TypeScript.

**Table 20 p-Values for post-hoc Wilcoxon signed rank test for RQ4 metrics between Rust and the other languages.**

| Metric | C | C | JavaScript | Python | TypeScript |
|---|---|---|---|---|---|
| $MI_{Original}$ | 0.2624 | 0.3308 | 0.2698 | 0.2624 | 0.2624 |
| $MI_{Visual\ Studio}$ | 0.2624 | 0.3308 | 0.2698 | 0.2624 | 0.2624 |

However, it is worth mentioning that several works in the literature from the latest years have highlighted the intrinsic limitations of the MI metric. A study by T. Kuipers underlines how the MI metric exposes limitations, particularly for systems built using object-oriented languages since it is based on the CC metric that will be largely influenced by small methods with small complexity; hence both will inevitably be low (*Kuipers & Visser, 2007*). *Counsell et al. (2015)* as well warn against the usage of MI for Object-Oriented software, highlighting the class size as a primary confounding factor for the interpretation of the MI metric. Several works have tackled the issue of adapting the MI to object-oriented code: *Kaur & Singh (2011)*, for instance, propose the utilization of package-level metrics. *Kaur, Kaur & Pathak (2014a)* have evaluated the correlation between the traditional MI metrics and the more recent maintainability metrics provided by the literature, like the CHANGE metric. They found that a very scarce correlation can be measured between MI and CHANGE. Lastly, many white and grey literature sources underline how different metrics for the MI can provide different estimations of the maintainability for the same code. This issue is reflected by our results. While the comparisons between different languages are mostly maintained by all three MI variations, it can be seen that all average values for original and SEI MI suggest very low code maintainability, while the average values for the Visual Studio MI would suggest high code maintainability for the same code artifacts.

## CONCLUSION AND FUTURE WORK

In this article, we have evaluated the complexity and maintainability of Rust code by using static metrics and compared the results on equivalent software artifacts written in C, C++, JavaScript, Python, and TypeScript. The main findings of our evaluation study are the following:

- The Rust language exhibited average verbosity between all considered languages, with lower verbosity than C and C++;
- The Rust language exhibited the most structured code organization of all considered languages. More specifically, the examined source code artifacts in Rust had a significantly higher number of arguments than most of the other languages;
- The Rust language exhibited average CC and values for Halstead metrics. Rust had a significantly lower COGNITIVE complexity with respect to all other considered languages;

- The Rust language exhibited average compound maintainability indexes. Comparative analyses showed that the maintainability indexes were slightly higher (hinting at better maintainability) than C and C++.

All the evidence collected in this paper suggests that the Rust language can produce less verbose, more organized, and readable code than C and C++, the languages to which it is more similar in terms of code structure and syntax. The difference in maintainability with these two languages was not significant. On the other hand, the Rust language provided lower maintainability than that measured for more sophisticated and high-level object-oriented languages.

It is worth underlining that the source artifacts written in the Rust language exhibited the lowest COGNITIVE complexity, meaning that the language can guarantee the highest understandability of source code compared to all others. Understandability is a fundamental feature of code during its evolution since it may significantly impact the required effort for maintaining and fixing it.

This work contributes to the existing literature of the field as a first, preliminary evaluation of static qualities related to maintainability for the Rust language and a first comparison with a set of other popular programming languages. As the prosecution of this work, we plan to perform further developments on the *rust-code-analysis* tool such that it can provide more metric computation features. At the present time, for instance, the tool is not capable of computing class-level metrics. However, it can only be employed to compute metrics only on function and class methods.

We also plan to implement parsers for more programming languages (e.g., Java) to enable additional comparisons. We also plan to extend our analysis to real projects composed of a significantly higher amount of code lines that embed different programming paradigms, such as the functional and concurrent ones. To this extent, we plan to mine software projects from open source libraries, e.g., GitHub.

### Funding
Mozilla Research funded this project with the research grant 2018 H2. The funders had no role in study design, data collection and analysis, decision to publish, or preparation of the manuscript.

### Grant Disclosures
The following grant information was disclosed by the authors:
Mozilla Research Fund: 2018 H2.

### Competing Interests
Luca Ardito is an Academic Editor for PeerJ Computer Science.
Luca Barbato is the owner of Luminem.

## Author Contributions

- Luca Ardito conceived and designed the experiments, authored or reviewed drafts of the paper, and approved the final draft.
- Luca Barbato performed the computation work, authored or reviewed drafts of the paper, and approved the final draft.
- Riccardo Coppola analyzed the data, prepared figures and/or tables, authored or reviewed drafts of the paper, and approved the final draft.
- Michele Valsesia performed the experiments, authored or reviewed drafts of the paper, and approved the final draft.

## Data Availability

Data is available at GitHub: https://github.com/SoftengPoliTo/rust-analysis.

Code is available at these three GitHub repositories:

– https://github.com/greensoftwarelab/Energy-Languages.
– https://github.com/SoftengPoliTo/SoftwareMetrics.
– https://github.com/SoftengPoliTo/rust-code-analysis.

## Supplemental Information

Supplemental information for this article can be found online at http://dx.doi.org/10.7717/peerj-cs.406#supplemental-information.

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
