# Peer review of "Evaluation of Rust code verbosity, understandability and complexity"

_PeerJ Computer Science, doi:10.7717/peerj-cs.406_

## Round 0.1 · original submission · Major Revisions

The three reviewers agree that the paper has some merits but they all raised relevant problems with the experimental design, namely the selection of the target applications and the lack of statistical analysis of the results. All reviewers provide several suggestions and remarks that should be taken into account in the revised version of the manuscript.

·

Basic reporting

The structure of the paper is appropriate, and the paper is easy to read and follow. The English language used in the paper is on par with only minor spelling mistakes, which can be mitigated with another round of proof-reading by the authors. The main contributions are appropriately highlighted in the introduction, making it easy for readers to determine an overview of the paper quickly.

-Please provide the reference for the statement in line 79 "...for example, software like Firefox, Dropbox, and Cloudflare use Rust."

-The manuscript is left-justified, not per the manuscript preparation instructions: "Left justify all text to the left margin. Do not 'full width' justify."

-The analysis for RQ1 should be supplemented with the reference that various lines of code metrics are the ones that are the most appropriate metrics for the programming language verbosity.

-On a similar note, what about the analysis classes for RQ2? Why only limit yourself to the methods?

-During the pipeline of the evaluation framework (Figure 1), please describe the differences of .json results after each step. It would be helpful to show an example of each JSON.

-It is described that compare.py is the main entry point of the source code files and results in the final .json with results. This is not evident from the pipeline in Figure 1 - here, compare.py is only one part of the whole pipeline, not the main script that runs other ones.

-How does tool rust-code-analysis analyze the C, C++, JavaScript, and other code? Is it not only meant for Rust code, as its name suggests? The quick review of the source code indicates that it analyses other languages, especially emphasizing this ability, so the readers do not get confused by its name.

-Be consistent with naming JSON (sometimes it appears as Json).

Experimental design

The experiment's design is appropriate, where various software metrics of the same software methods implemented in different languages are compared among themselves. The data used in the experiment and the source code used in the analysis are both publicly available. This makes the experiment repeatable.

-The main issue with the experiment is the lack of any statistical analysis of the results. There is enough data (enough analyzed software files) to make at least a basic statistical comparison. PeerJ CS is a high impacting journal, and thus, the methodology should be suitable. One could use ANOVA for repeated measurements or Friedman's ANOVA on every metric to find if the differences shown in the charts and table are due to chance (source of software or programmer making them) or they are statistically significant. If there are differences, use posthoc tests (i.e., Wilcoxon signed-rank with correction for multiple comparisons Holm-Bonferroni) to determine the real answers for posed research questions.

-Provide reasoning on why some of the programming languages were used in the experiment and others were not. Yes, it is mentioned that the implemented module only supports some of them, but why those. Are the chosen programming languages valid alternatives in some information systems (web assembly programming, etc.)? With this, you will introduce readers to Rust's typical applications and show what its main alternatives are.

Validity of the findings

The main issue with the findings was already mentioned in part about the experimental design - the lack of any in depath analysis of the results. Additionally, several other issues have to be addressed.

-The discussion on the differences of LLOC is a bit too narrow. Authors only explain the differences in this metric as the product of more types of logical statements available in Rust. In this case, Python would have the lowest LLOC count, which is not evident from the results. Also, the sheer number of available types does probably not correlate with the higher usage of those. If you argue that it does, please provide a reference or at least a viable justification. I would argue that there could be other reasons for more LLOC, which are fundamentally simple. For example, it could indicate that the logical statements are more elementary (do less in one call) than in other languages, so more are needed. Does this make code more verbose? Probably, but it's open for discussion. What if this is the key reason why the cyclomatic complexity and cognitive complexity are the highest with Rust?

-On a similar note, the number of methods and sum of arguments discrepancies could probably further explain Rust's method arguments' lack of default values.

-Also, is the higher count of methods the sign of better structure of the code? The answer to RQ2 suggests this. Again, this implies that using the one method without its variants (for different argument count) is superior. Please elaborate on this point.

Additional comments

The paper presents the results of an analysis of maintainability metrics and other software metrics software written in the programming language Rust. The authors took a publicly available repository of the different procedures written in various programming languages and compared them. The paper's main findings are lenient to the Rust programming language, as Rust results as the language in which software is written without too much complexity, is verbose, and is not too hard to maintain. The paper is derived from the final thesis, which (after my review and search) has never been published before.

Reviewer 2 ·

Basic reporting

- In section Introduction, the maintainability characteristic should be linked to well-known software quality standards such as ISO 9126, ISO 25010; they also provide the sub-characteristics for maintainability, allowing for a finer grained approach
- Introduction section could also use more recent references, as there exists a lot of post-2017 research on the topic.
- Rephrase line 46-47 to eliminate repetition
- Revise reference on line 101
- Line 125: "open-source algorithms" - perhaps this needs a bit of clarification; are they open-source algorithms or are the algorithms implemented in open-source code?
- Table 4 - the meaning for some of the formula terms (column Formula, N1 and N2 for example) remains unclear.
- Perhaps a reorganisation of the Tables on pages 4 - 6 would improve the paper's readability, as currently there is a 1.5 page gap in the article text.
- "NARGS and NEXITS are two software metrics defined by Mozilla and have no equivalent in the literature about maintainability metrics". In that case, what makes the authors employ these metrics for studying the maintainability characteristic?
- Lines 155-156 please recheck
- Figure 1, first two boxes (labeled 'Source code' and the one with file extensions) should be merged, as they make the idea clear; Also, perhaps it would be better to eliminate the .json boxes and represent the entire process on a single line; perhaps use 'JSON' as an annotation over the arrows, to show that was the selected format for data transfer.
- Lines 199-201 - it's not clear to me what this paragraph refers to; perhaps its intent could be further clarified by the authors
- Listing 1 does not improve the quality or understandability of the article; perhaps it would be best to include in the repository's GitHub readme and direct the reader to that using a suitable footnote.
- Lines 403-404, 435-436 refer the wrong Tables/Figures.
- Lines 437-438 - the temporal characteristic of the MI is not clear; changes in its value could be interpreted as a modification of maintainability, but the metric itself reports a singular value.

Experimental design

- The Paper should be structured according to existing best practices regarding case study research (e.g: Runeson and Host - Guidelines for conducting and reporting case study research in software engineering)
- The Maintainability Index was first elaborated for a number of C systems, and has come under strong criticism recently for not being able to adequately express the maintainability characteristic in newer paradigms (such as object oriented) and newer programming languages. While there is still merit in using it, authors should address the existence of relevant concerns. In addition, further explanation is required regarding the different forms employed for the MI. This, together with the selection of rather simple metrics to assess maintainability raises issues regarding the accuracy of the authors' measurements and their validity.
- I am not convinced that RQ1 - RQ3 are related to software maintainability, as it is understood from a software engineering perspective.
- I believe authors should drill down and present a comparative evaluation at target application level; do the descriptive statistics presented hold at each application, or are there more interesting findings?

Validity of the findings

- The selection of the 9 algorithms is arbitrary, and introduces an important threat to the external validity of the study. In addition, it is usually the case that algorithm implementations are but a small part of most large-scale systems, so it is not at all clear how the maintainability characteristic that was evaluated using these algorithm implementations will scale upwards.
- A further threat is represented by the fact that the studied algorithms were implemented as part of a software suite to study the performance of different programming languages/runtimes. This could have a further effect on the representativeness of these code bases for larger scale applications developed using those languages.
- With regards to RQ1, authors did not detail the relation between code verbosity and maintainability. Existing methodologies to determine maintainability, and at a higher level than the MI, such as technical debt are concerned with existing best practices, detection of code smells and other weaknesses; as such, it is unclear how the innate verbosity of a language will translate to the maintainability characteristic.
- Regarding the authors' answer to RQ2, the discussion should be based on the implementation of larger-scale software; it should also include a discussion on the source code author(s) programming style, as that can have an impact on these complexity metrics, especially when considering such a limited code base. This is true especially in the case of the NARGS and NEXITS metrics that are not extensively studied in the literature.
- The application of the Halstead time and bugs metrics to a new programming language/construct introduces further threats to validity; these proposed values (division by 18 and 3000, respectively) should most likely be evaluated empirically first. This is partly addressed by the authors in the Threats to Validity section.

Additional comments

The paper is competently written and approaches a subject of current interest in research. However, I believe that the title is out of sync with the paper's contents. The selection of target applications is severely limited, and suitable for an introductory, or position paper on the subject, and not a full journal publication. Furthermore, the selection of metrics to assess maintainability is limited to simplistic measurements. Recent research into maintainability generally employs more complex measures such as technical debt or the impact of code smells (such as measured using SonarQube or Ptidej). Of course, their application assumes a larger target code base to provide meaningful results. Relating to the selection of target applications, the 9 implementations are part of a benchmarking suite, and as such introduce an important threat related to the validity of the conclusions, when these are externalized to other kinds of software (e.g. open-source world or proprietary implementations of large-scale systems).

I believe that in order to work well, the paper should be re-targeted towards examining source code verbosity and understandability across different languages. In this way, the selection of target applications gains relevance, and well-known metrics such as the Halstead suite (that are no longer used to evaluate maintainability) can be more successfully employed.

Reviewer 3 ·

Basic reporting

Authors have followed the professional article structure and shared the raw data. I commend the authors for their work but certain issues need to be resolved before acceptance.

1. Mapping of figures/tables should be thoroughly cross checked with the places they are referenced in the manuscript. Authors need to correct the table and figure referencing
For example,.
In 358, “In the table, we report the mean and…….” Which table authors are referring to?
Line 403 “The boxplots in Figure 4 and Table 9 report the distributions, mean, and median of the Halstead….” Authors have cited wrong table and figure reference. It should be Figure 5 and Table 10.
Similar observation is made at line 435-436. “The boxplots in Figure 4 and Table 9 report the distributions, mean, and median of the Maintainability Indexes computed for the six different programming languages.”. This is repeated line with wrong references.

2. he authors use "we" too much in the paper, while I suggest to use "the paper".

3. Different notations are used for same object. It is recommended that authors should use single term. Some examples are;
Json, json, .json;
line 321 SLOC line 322 souce loc. In line 323 and 324 lines of code. C
OGNITIVE complexity (line 375, 376) or Cognitive complexity (line 390, 395, 396), or cognitive complexity (line 393) or Cognitive Complexity(396).
Program difficulty ( line 406, 409) and Difficulty(line 408)

4. As authors has mentioned in line 438, “Halstead Volume (V), the Cyclomatic Complexity (CC),…”, they must mention the acronyms for all other terms when first used in paper.

5. The paper is well organized. But at some points, restructuring of sentences is required. Few examples are:

Line 269-271: Multiple use of and in one sentence. “Concerning the original implementation of the rust-code-analysis tool, we have forked the project and performed modifications on it by adding metrics computations (e.g., the COGNITIVE metric) and changes to the possible output format provided by the tool.”

Line 440-441: “By using all the formulas for the Maintainability Index, we computed for the source files written in Rust an average MI that placed the fourth among all considered programming languages."
“This very low value of the cognitive per method for Rust is related……..” should be “This very low value of the cognitive complexity per method for Rust is related……..”

6. Minor grammatical errors were exposed. For example,
In 335 “….with the second-highest, mean being 59 for the…..” should be “….with the second-highest mean being 59 for the…..”

In captions of Figure 1, Figure 2 :” Distributions of the metrics about…..” should be “Distribution of the metrics about…”.
Line 95: Systematic Literature review should be systematic literature review.

7. Please check the PeerJ reference format and references should be consistent with that format.
The references in manuscript do not follow a commat format. For example,
Alqadi, B. S. and Maletic, J. I. (2020). Slice-based cognitive complexity metrics for defect prediction. In 2020 IEEE 27th International Conference on Software Analysis, Evolution and Reengineering (SANER), pages 411–422. IEEE.
Astrauskas, V., Mu¨ller, P., Poli, F., and Summers, A. J. (2019). Leveraging rust types for modular specification and verification. Proceedings of the ACM on Programming Languages, 3(OOPSLA):1– 30.

Experimental design

The experiments were well implemented, and the results are consistent. Work is novel. A tool is constructed to extract metrics of Rust and object-oriented languages. Metrics are collected for 9 program codes written in 6 programming languages. The paper is well written, the structure makes it easy to follow. Research questions are well formulated.

1. I would request authors to comment on their selection of metrics to be extracted from code. Why they did not extract object-oriented metrics?

2. Algorithms are language independent. Authors have use codes of different languages to do comparative analysis. In Table 6 title, algorithms should be replaced by code. Similarly, in complete text, whenever referring to code, replace ‘algorithm’ by ‘code’.

3. Authors mentioned and analyzed maintainability index in subsection 4.4. I would suggest authors to include some range of maintainability index (for example: bad, average, good, acceptable). This will give more clarity to readers about its relevance.

4. In Table 5, authors have scribed the three variants of MI metric. It is suggested to add reference and little detail for each definition in corresponding section.

5. Table 1 shows that CKJM extracts JAVA and C metrics.
But CKJM collects metrics only for compiled JAVA classes. CKJM stands for Chidamber and Kemerer Java Metrics. It does not work for C code. Authors need to rectify it.
Spinellis D. Tool writing: a forgotten art?(software tools). IEEE Software. 2005 Jul 11;22(4):9-11

Validity of the findings

I appreciate authors to provide all underlying data supporting the replication of the work.

1. In results section, conclusions are well stated for each RQ. But comparative analysis need to be further strengthened by using statistical tests. Authors must include statistical validation of their results. Depending on the nature of data, they can use either parametric or non-parametric tests to statistically validate the results.

2. Conclusion section need to be elaborated. Authors should include main contributions in it.

---

## Round 0.2 · Minor Revisions

I am glad to inform you that pending MINOR revisions your paper will be accepted. The reviewers are satisfied with the previous revision and pinpointed only a few issues that should be addressed in this revision.

·

Basic reporting

The authors significantly improved the paper in accordance with my (and other reviewer's) comments. As I see no other issues left uncovered, I have no further objections.

Experimental design

The authors included the statistical analysis, as per my suggestion, and addressed all of the raised issues.

Validity of the findings

The additional explanations make the newest version of the paper higher in quality, thus the validity of the findings is not questionable anymore.

Additional comments

The authors significantly improved the paper in accordance with my (and other reviewer's) comments. As I see no other issues left uncovered, I have no further objections.

Reviewer 2 ·

Basic reporting

• Line 127 - MetricsReloaded seems to be doing fine as an IntelliJ plugin, last release was December 2020 (https://plugins.jetbrains.com/plugin/93-metricsreloaded)
• In Section 3.1, there is the possibility of detailing each RQ immediately after it was stated; this might further help readers maintain the current context and better understand the scope of the research; however, this is just a suggestion and deferred to the authors' judgement
• Line 218 - "speed of programming languages" might not best express the authors' intention; what is actually measured is how quickly code written in a given language can be executed after being translated/interpreted by the compiler/interpreter target architecture, as a matter of optimization.
• Line 239 - capitalize 'Figure'
• Lines 249 and 266 look a weird, as they start with a non-capital letter, and it's unclear whether by intention or the result of a formatting error.
• Line 272 should mention that Listing 1 is in the annex, as it's many pages away. Also, the listings might be put into a table with 2 columns and only take up 2 pages for brevity, or included in a data replication package on figshare/zenodo with their own ISBN.
* Line 493 'Whe' (typo)

Experimental design

no comment

Validity of the findings

no comment

Additional comments

The paper was revised according to the findings and suggestions resulting from the first review. The current version provides much more detail regarding the paper's main objective, the means employed to fulfil the stated objective as well as the methodology and threats to validity. Observations resulting from a careful reading can be found in the section 'Basic Reporting'.

Reviewer 3 ·

Basic reporting

I appreciate authors for incorporating the suggested changes in the manuscript.

1. Minor grammatical mistakes are still in the manuscript. Please check use of commas, articles, correct preposition usage. I have listed only few examples for your reference. Kindly crosscheck complete manuscript for correct grammar usage.

Line 48: "metrics under the categories of Size, Coupling, Complexity and Inheritance [7]." should be "metrics under the categories of Size, Coupling, Complexity, and Inheritance [7]."


Line 88: "ownership model to guarantee memory-safety and thread-safety; productivity, with integrated package" should be "ownership model to guarantee memory-safety and thread-safety; productivity, with an integrated package".


Line 109: "review, it is found that the following set of open-source tools is able to cover most of quality metrics" should be "review, it is found that the following set of open-source tools can cover most of quality metrics".

Line 144: "interpreting the results from developers and researchers standpoint." I think you missed to add apostrophe's in developers and researchers.

Line: 171 "lines or comments; the count, however, depends on the physical format of the statements and on programming". Omit "on" before programming. Repetitive use of On.

Line 223: "Table 7 lists the code artifacts used (sorted out alphabetically) and provides a brief description for each" should be "Table 7 lists the code artifacts used (sorted out alphabetically) and provides a brief description of each"


2. More than one references can be clubbed together.
for example:
Line 52: "code metrics to predict or infer the maintainability of a project [9], [10], [11]." may be rewritten as "code metrics to predict or infer the maintainability of a project [9-11]."

3. Earlier also, I suggested to go through all the references. Kindly comply with PeerJ reference format. Below is only one example, you need to recheck all references.

Line 855 17] Abhiram Balasubramanian, Marek S Baranowski, Anton Burtsev, Aurojit Panda, Zvonimir Rakamaric´, and Leonid Ryzhyk. System programming in rust: Beyond safety. In Proceedings of the 16th Workshop on Hot Topics in Operating Systems, pages 156–161, 2017.
Line 858 [18] Vytautas Astrauskas, Peter Mu¨ller, Federico Poli, and Alexander J Summers. Leveraging rust types for modular specification and verification. Proceedings of the ACM on Programming Languages, 3(OOPSLA):1–30, 2019.

Experimental design

Authors have justified the work by revising the manuscript.

1. Statistical tests are now included . My concern is that
a) please state the hypothesis set for conducting the statistical tests.
b) if Wilcoxon test is used to compare multiple pairs, you need to use a p-value correction like a Bonferroni correction for the results to be reliable.

Validity of the findings

Work is novel. Good work done by the authors. Conclusion and future work is now in better shape. Supporting data is provided and conclusions are linked with original research questions.

Kindly modify Wilcoxon test with Bonferroni correction for reliability of the results, as I stated in experimental design.

---

## Author Rebuttal · Round 0.2

Dear Editors and Reviewers of PeerJ - Computer Science,

Please find hereby enclosed the revision of the paper:
"Evaluation of Rust code complexity and maintainability"

We have modified the manuscript following the major reviews that were received.

In summary, we have performed the following modifications to the manuscript:

- according to one of the main comments of reviewer 2, we have changed the title and the focus of the manuscript to avoid using the measurements for the first research questions as direct measures of maintainability. Therefore, the new title of the article is "Evaluation of Rust Code Verbosity, Understandability and Complexity";
- we have added statistical analyses for all the metrics we measured and we reported them in the manuscript;
- we have added details about the rust-code-analysis tool that we employed and intermediate results. Please note that conflicting comments were raised by two reviewers about this point: rev. 1 asked for an example for each intermediate JSON file produced, while rev. 2 asked for the removal of the examples from the manuscript. We opted for the inclusion of the examples as an appendix in the current revision. We are still open to moving the examples to an external online appendix;
- we have restructured the manuscript in order to follow recent and established guidelines for reporting case studies;
- we have added additional critical discussion about the MI, NARGS, and NEXITS metrics;
- we have added the threats to validity that were signaled by the reviewers;
- we have added an analysis of the outliers of the distributions and motivated them by analyzing the related software code artifacts;
- we have performed an additional round of grammar check and proof-reading of the manuscript and fixed all the signaled inconsistencies and typos.

We hope that we have addressed the issues raised by the reviewers in the best way possible. We thank the reviewers for their constructive and detailed insight that helped us enrich our work significantly.

Below we report the comments from the review, with our point-by-point responses in blue.

Sincerely,

Luca Ardito,
Luca Barbato,
Riccardo Coppola,
Michele Valsesia

**Reviewer 1**

-Please provide the reference for the statement in line 79 "...for example, software like Firefox, Dropbox, and Cloudflare use Rust."

-The manuscript is left-justified, not per the manuscript preparation instructions: "Left justify all text to the left margin. Do not 'full width' justify."

-The analysis for RQ1 should be supplemented with the reference that various lines of code metrics are the ones that are the most appropriate metrics for the programming language verbosity.

> We have motivated the selection of multiple measurements for the LOCS of a source file with the following paragraph:
>
> *The rationale behind using multiple measurements for the lines of code can be motivated by the need for measuring different facets of the size of code artifacts and of the relevance and content of the lines of code. The measurement of physical lines of code (PLOC) does not take into consideration blank lines or comments; the count however depends on the physical format of the statements and on programming style, since multiple PLOC can concur to form a single logical statement of the source code. PLOC are sensitive to logically irrelevant formatting and style conventions, while LLOC are less sensitive to these aspects [33]. In addition to that, the CLOC and BLANK measurements allow a finer analysis of the amount of documentation (in terms of used APIs, and explanation of complex parts of algorithms) and formatting of a source file.*

-On a similar note, what about the analysis classes for RQ2? Why only limit yourself to the methods?

> Rust-code-analyis computes metrics on both functions and class methods. Currently, rust-code-analysis does not implement any class metric, so they haven't been considered in our analysis. We consider to increase the number of metrics of the RCA tool in our future work, and we have now made it explicit in the Conclusion and Future Work section.
>
> *As the prosecution of this work, we plan to perform further developments on the rust-code-analysis tool such that it can provide more metric computation features. At the present time, for instance, the tool is not capable of computing class-level and object-oriented metrics, but it can only be employed to compute metrics only on function and class methods.*

-During the pipeline of the evaluation framework (Figure 1), please describe the differences of .json results after each step. It would be helpful to show an example of each JSON.

We have added excerpts for the three JSON files produced at each step of the framework. For readability reasons, we could not insert the full files inside the text of the paper, but we added a footnote with a link to the folder of the GitHub project where the JSON files are generated

-It is described that compare.py is the main entry point of the source code files and results in the final .json with results. This is not evident from the pipeline in Figure 1 - here, compare.py is only one part of the whole pipeline, not the main script that runs other ones.

The first diagram in Figure 1 represents the dataflow pipeline, thus how data are computed and passed among processes. We have added a second diagram, in figure 2, where we represent the process stack, whose goal is to determine how the processes are launched. In this chart, the lowest block of the stack is the entry point, identified by a specific color, while the other blocks are the relative subprocesses.

We have updated the description of the execution of the framework as follows:

*A graphic overview of the framework is provided in Figure 1. The diagram only represents the logical flow of the data in our framework since the actual flow of operations is reversed, being the \emph{compare.py} script the entry point of the whole computation.% as described later in this section.*

*The rust-code-analysis tool is used to compute static metrics and save them in the JSON format. The analyzer.py script receives as input the results in JSON format provided by the rust-code-analysis tool, and format them in a common notation, which is more focused on academic facets of the computed metrics, rather than the production ones used by the rust-code-analysis default formatting. The compare.py has been developed to call the analyzer.py script and to use its results to perform pair-by-pair comparisons between the JSON files obtained for source files written in different programming languages. These comparison files allow us to immediately assess the differences in the metrics computed by the different programming languages on the same software artifacts. The stack of commands that are called in the described evaluation framework is shown in figure 2.*

-How does tool rust-code-analysis analyze the C, C++, JavaScript, and other code? Is it not only meant for Rust code, as its name suggests? The quick review of the source code indicates that it analyses other languages, especially emphasizing this ability, so the readers do not get confused by its name.

The name `rust-code-analysis` refers to the innovative possibility to compute static metrics using a program written in Rust programming language. To parse all the considered languages, it has been adopted a specific Rust library called tree-sitter that receives a source code as input and produces in the output the relative AST. rust-code-analysis, from the AST, extracts the static metrics. We have added details about how rust-code-analysis works, in section 3.3.1. We report in the following the excerpt from the manuscript:

*rust-code-analysis builds, through the use of an open-source library called tree-sitter, builds an Abstract Syntax Tree (AST) to represent the syntactic structure of a source file. An AST differs from a Concrete Syntax Tree because it does not include information about the source code less important details, like punctuation and parentheses.*

*On top of the generated AST, rust-code-analysis performs a division of the source code in spaces, i.e. any structure that can incorporate a function. It contains a series of fields such as the name of the structure, the relative line start, line end, kind, and a \emph{metric} object, which is composed of the values of the available metrics computed by rust-code-analysis on the functions contained in that space. All metrics computed at the function level are then merged at the parent space level, and this procedure continues until the space representing the entire source file is reached. The tool is provided with parser modules that are able to construct the AST (and then to compute the metrics) for a set of languages: C, C++, C#, Go, JavaScript, Python, Rust, Typescript. The programming languages currently implemented in rust-code-analysis have been chosen because they are the ones that compose the mozilla-central repository, which contains the code of the Firefox browser. The metrics can be computed for each language of this repository with the exception of Java, which does not have an implementation yet, and HTML and CSS are excluded because they are formatting languages.*

*rust-code-analysis can receive either single files or entire directories, detect whether they contain any code written in one of its supported languages, and output the resultant static metrics in various formats: textual, JSON, YAML, toml, cbor.*

-Be consistent with naming JSON (sometimes it appears as Json).

The experiment's design is appropriate, where various software metrics of the same software methods implemented in different languages are compared among themselves. The data used in the experiment and the source code used in the analysis are both publicly available. This makes the experiment repeatable.

-The main issue with the experiment is the lack of any statistical analysis of the results. There is enough data (enough analyzed software files) to make at least a basic statistical comparison. PeerJ CS is a high impacting journal, and thus, the methodology should be suitable. One could use ANOVA for repeated measurements or Friedman's ANOVA on every metric to find if the differences shown in the charts and table are due to chance (source of software or programmer making them) or they are statistically significant. If there are differences, use posthoc tests (i.e., Wilcoxon signed-rank with correction for multiple comparisons Holm-Bonferroni) to determine the real answers for posed research questions.

We have applied a non-parametric Kruskal-Wallis test to identify statistically significant differences among the different sets of metrics for each language.

For significantly different distributions we have finally applied post-hoc comparisons with Wilcoxon signed rank sum test to analyze the difference between the metrics measured for Rust and the other five languages in the set.

-Provide reasoning on why some of the programming languages were used in the experiment and others were not. Yes, it is mentioned that the implemented module only supports some of them, but why those. Are the chosen programming languages valid alternatives in some information systems (web assembly programming, etc.)? With this, you will introduce readers to Rust's typical applications and show what its main alternatives are.

> We have added the rationale for the selection of the 5 languages for the metric computation:
>
> *We were restricted to a limited number of 5 programming languages for the comparison because those languages (additional details are provided in the next section) were the common ones for the Energy-Languages repository and the set of languages that are correctly parsed by the tooling we employed in the experiment conduction.*
>
> *[...]*
>
> *The tool is provided with parser modules that are able to construct the AST (and then to compute the metrics) for a set of languages: C, C++, C#, Go, JavaScript, Python, Rust, Typescript. The programming languages currently implemented in rust-code-analysis have been chosen because they are the ones that compose the mozilla-central repository, which contains the code of the Firefox browser. The metrics can be computed for each language of this repository with the exception of Java, which does not have an implementation yet, and HTML and CSS are excluded because they are formatting languages.*

The main issue with the findings was already mentioned in part about the experimental design - the lack of any in depath analysis of the results. Additionally, several other issues have to be addressed.
-The discussion on the differences of LLOC is a bit too narrow. Authors only explain the differences in this metric as the product of more types of logical statements available in Rust. In this case, Python would have the lowest LLOC count, which is not evident from the results. Also, the sheer number of available types does probably not correlate with the higher usage of those. If you argue that it does, please provide a reference or at least a viable justification. I would argue that there could be other reasons for more LLOC, which are fundamentally simple. For example, it could indicate that the logical statements are more elementary (do less in one call) than in other languages, so more are needed. Does this make code more verbose? Probably, but it's open for discussion. What if this is the key reason why the cyclomatic complexity and cognitive complexity are the highest with Rust?

> After the first round of review, we have examined the rust-code-analysis tool to better understand the motivations behind such a high value for the LLOC metric measured on Rust source codes. We found that there were defects in the components of the RCA tool that were involved in the computation of the LLOC metric. Specifically, the tool counted as statements some nodes of the abstract syntax tree that were not actual statements. This erroneous classification of the nodes led to exceptional

growth in the counted number of statements. We thoroughly analyzed files in input and the results and implemented a fix on the rust-code-analysis repository. We added this fix in a pull request that we sent to the repository curators and the current version of the repository that we are considering is v0.1.0 (https://github.com/SoftengPoliTo/SoftwareMetrics/releases/tag/v0.0.1).

After fixing the module for the computation of the tool, we have recomputed all the LLOC measures. The new counts provide an LLOC result that is in line with those measured for the other 5 languages. These results also invalidate the reasoning that we had provided about the LLOC metric in the previous version of the manuscript. We have updated the reasoning about the LLOC incorporating the suggestions provided by the reviewer, and providing as a side note (for the comparison between Rust and C / C++) the point about the number of different types of statements offered by the language.

We have added reasoning about the possible connection between the LLOC and the complexity in the discussion about the latter. The Rust language indeed had the lowest CC and COGNITIVE between all the languages.

We report in the following the updated paragraph containing the discussion about the LLOC metric:

*A slightly different trend is assumed by the Logical Lines of Code (LLOC) metric (i.e., the number of instructions or statements in a file). In this case, the mean number of statements for Rust code is higher than the ones measured for C, C++, and TypeScript, while the SLOC and PLOC metrics are lower. The Rust scripts also had the highest median LLOC. This result may be influenced by the different number of types of statements that are offered by the language. For instance, the Rust language provides 19 types of statements while C offers just 14 types (e.g., the Rust statements \emph{If let} and \emph{While let} are not present in C). The higher amount of logical statements may indeed hint at a higher decomposition of the instructions of the source code into more statements, i.e., more specialized statements covering fewer operations.*

-On a similar note, the number of methods and sum of arguments discrepancies could probably further explain Rust's method arguments' lack of default values.

We agree with the reviewer with the justification about the highest argument number (total) caused by the missing possibility of having default values in the Rust language. This characteristic may thereby lead to multiple variations of the same method to take into account changes in the parameter, hence leading to a higher sum of the number of arguments (and not to a higher average of the number of arguments). We have added this clarification in the section of the manuscript related to RQ2.

-Also, is the higher count of methods the sign of better structure of the code? The answer to RQ2 suggests this. Again, this implies that using the one method without its variants (for different argument count) is superior. Please elaborate on this point.

> As a preliminary clarification of the answer to RQ2, with the "most structured" expression, we just report the fact that the Rust code is more divided in different methods than the other languages.
>
> Our explanation of the gathered numbers is the following. The usage of more methods means a more structured code, which translates to more arguments if the sum of all arguments of the methods is considered (so higher NOM and NARGS_SUM measurements), whilst leading to smaller average NARGS. Having more methods can result in higher verbosity of the code, and verbosity may have a negative impact on code maintainability because it may render it difficult for the reader to identify which portions of the code are in charge of carrying specific operations (this topic is also tackled below in another response to the same reviewer).
>
> This result, however, is balanced in our measured set of metrics by the COGNITIVE value for the Rust language, which is the lowest among all languages. This result, albeit preliminary and to be verified on bigger and more numerous software projects, may suggest that the inherent characteristics of the Rust language would be an aid to have a better readability and maintainability of software artifacts even in the presence of higher NOM and CC. We have inserted this possible interpretation of the results in the section answering RQ3.

Comments for the Author
The paper presents the results of an analysis of maintainability metrics and other software metrics software written in the programming language Rust. The authors took a publicly available repository of the different procedures written in various programming languages and compared them. The paper's main findings are lenient to the Rust programming language, as Rust results as the language in which software is written without too much complexity, is verbose, and is not too hard to maintain. The paper is derived from the final thesis, which (after my review and search) has never been published before.

**Reviewer 2**

- In section Introduction, the maintainability characteristic should be linked to well-known software quality standards such as ISO 9126, ISO 25010; they also provide the sub-characteristics for maintainability, allowing for a finer grained approach

> Definitions have been mutuated from the mentioned standards and references to them have been added in the introduction section

- Introduction section could also use more recent references, as there exists a lot of post-2017 research on the topic.

> We have added 7 references to post-2017 research in the introduction section

- Rephrase line 46-47 to eliminate repetition
- Revise reference on line 101

- Line 125: "open-source algorithms" - perhaps this needs a bit of clarification; are they open-source algorithms or are the algorithms implemented in open-source code?

> With "open-source algorithms" we actually wanted to refer to the implementation of known algorithms in code (in various languages) that was part of an open-source repository. We have clarified this aspect at line 125 and removed the ambiguity between "algorithm" and "code" throughout the remainder of the manuscript.

- Table 4 - the meaning for some of the formula terms (column Formula, N1 and N2 for example) remains unclear.

> We have added into the table the four "base measures" that are the basis for the computation of all the remaining Halstead metrics

- Perhaps a reorganisation of the Tables on pages 4 - 6 would improve the paper's readability, as currently there is a 1.5 page gap in the article text.

> We have re-arranged the tables as suggested by the reviewer.

- "NARGS and NEXITS are two software metrics defined by Mozilla and have no equivalent in the literature about maintainability metrics". In that case, what makes the authors employ these metrics for studying the maintainability characteristic?

> We discussed with the original developers of the metrics at Mozilla. The main rationale behind the hypothesized connection between NARGS, NEXITS, and maintainability is the following:

- a function with a high number of arguments can have higher complexity in being analyzed because it will likely have more paths of execution depending on the values of the arguments. it will also be harder to unit test

- a function with many exits may include higher complexity in reading the code for performing maintenance efforts. This also builds on the traditional best practice for the maintainability of single-entry single-exit functions.

We agree with the reviewer, however, that the metrics are still not objects of empirical validation on large codebases and so they cannot be directly correlated to maintainability without specifying that they are at most not-validated proxies of that property. However, they are extensively used in production in mozilla-central, a very large open-source codebase.

We have therefore toned down all the conclusions about maintainability based on NEXITS and NARGS and interpreted them as they are, i.e. measurements of the organization and complexity of the code. We have also added a threat to the internal validity related to the missing empirical validation of the two metrics in the specific section. Finally, we have also described the intuition on which the metrics are based when we first introduce them.

- Lines 155-156 please recheck

- Figure 1, first two boxes (labeled 'Source code' and the one with file extensions) should be merged, as they make the idea clear; Also, perhaps it would be better to eliminate the .json boxes and represent the entire process on a single line; perhaps use 'JSON' as an annotation over the arrows, to show that was the selected format for data transfer.

The source code label has been removed and file extensions have been merged. As requested, the dataflow chart shows all processes along a single line. In addition, JSON boxes have been moved over the arrows that connect the processes. To better discriminate between files and processes, blue color has been assigned to the former, while a red one to the latter. To better clarify the process flow in addition to the data flow, we have added a stack representation of the called processes in figure 2.

- Lines 199-201 - it's not clear to me what this paragraph refers to; perhaps its intent could be further clarified by the authors

The paragraph was badly placed after previous modifications on the paper. We have moved it to the right placement after we mentioned the tool that we adopted and extended. The meaning of the paragraph was indeed to motivate the selection of such a tool in the Rust language.

- Listing 1 does not improve the quality or understandability of the article; perhaps it would be best to include in the repository's GitHub readme and direct the reader to that using a suitable footnote.

We have added footnotes linking to the repository readme and to the folders where the results of the scripts are generated. In answer to a comment by reviewer 1, we added as well examples of the JSON files provided by each step of the framework

- Lines 403-404, 435-436 refer the wrong Tables/Figures.

- Lines 437-438 - the temporal characteristic of the MI is not clear; changes in its value could be interpreted as a modification of maintainability, but the metric itself reports a singular value.

We clarified the definition of the aim of maintainability index in a clearer way, avoiding to hint at MI taking into consideration information about the temporal evolution of the software.

Therefore, we changed "an estimate of software maintainability over time"

To "an estimate of how maintainable (easy to support and change) the source code is"

- The Paper should be structured according to existing best practices regarding case study research (e.g: Runeson and Host - Guidelines for conducting and reporting case study research in software engineering)

We have re-structured the paper according to existing best practices for documentation of case studies.

Specifically, we have substituted the GQM template with Robson's template for case study description (table 2), and we have structured the study design section of the manuscript according to the reporting structure by Jedlitschka and Pfahl.

- The Maintainability Index was first elaborated for a number of C systems, and has come under strong criticism recently for not being able to adequately express the maintainability characteristic in newer paradigms (such as object oriented) and newer programming languages. While there is still merit in using it, authors should address the existence of relevant concerns. In addition, further explanation is required regarding the different forms employed for the MI. This, together with the selection of rather simple metrics to assess maintainability raises issues regarding the accuracy of the authors' measurements and their validity.

We have added an additional explanation of the different formulas used for the computation of MI, as well as a discussion of the theoretical maintainability ranges and how they vary according to the variant of the MI formula.

We have added at the end of the related discussion section a discussion of the possible issues of MI as already investigated in the literature and a brief critical evaluation of the results that we got in our experiment. We report this part of the manuscript in the following:

*It is worth however mentioning that several works in the literature from the latest years have highlighted the intrinsic limitations of the MI metric. A study by T. Kuipers underlines how the MI metric exposes limitations particularly for systems built using object-oriented languages, since it is based on the CC metric that will be largely influenced by small methods with small complexity, and will inevitably be low. Counsell et al. as well, warn against the usage of MI for OO software, highlighting the class size as a primary confounding factor for the interpretation of the MI metric. Several works have tackled the issue of adapting the MI to object-oriented code: Kaur et al., for instance, propose the utilization of package-level metrics. Kaur et al. have evaluated the correlation between the traditional MI metrics and more recent maintainability metrics provided by the literature, like the CHANGE metric. They found that a very scarce correlation can be measured between MI and CHANGE. Lastly, many white and grey literature sources underline how different metrics for the MI can provide different estimations of the maintainability for the same code. This issue is reflected by our results. While the comparisons between different languages are mostly maintained by all three MI variations, it can be seen that all average values for original and SEI MI suggest very low code maintainability, while the average values for the Visual Studio MI would suggest high code maintainability for the same code artifacts.*

- I believe authors should drill down and present a comparative evaluation at target application level; do the descriptive statistics presented hold at each application, or are there more interesting findings?

We have performed an analysis of the outliers to identify specific aspects of the studied source code artifacts, and we have added details at the end of each subsection of the Discussion chapter, regarding the different metrics.

- The selection of the 9 algorithms is arbitrary, and introduces an important threat to the external validity of the study. In addition, it is usually the case that algorithm implementations are but a small part of most large-scale systems, so it is not at all clear how the maintainability characteristic that was evaluated using these algorithm implementations will scale upwards.

We agree with the reviewer that an important threat to validity is introduced by the selected 9 algorithms. As we have reported also in the threats to validity section of the new revision of the manuscript, our original idea was to evaluate our set of metrics on larger-scale projects. However, this proved unfeasible because it was not possible for us to find a set of larger projects that were translated into all of the considered languages. In future extensions, we may consider adopting one or more medium-to-large projects written in at least two of the selected languages. In the present work, we scaled down our evaluation and resort to selecting a set of small algorithms that were already available in all of the languages with which the rust-code-analysis is compatible. We are aware that the applicability of the findings of this manuscript to bigger projects is uncertain, and we have stressed more the concept in the threats section. We believe however that our evaluation of small

source artifacts can properly serve as a preliminary investigation of the verbosity, organization, and readability of the selected set of languages.

- A further threat is represented by the fact that the studied algorithms were implemented as part of a software suite to study the performance of different programming languages/runtimes. This could have a further effect on the representativeness of these code bases for larger scale applications developed using those languages.

We agree with the reviewer that the selection of the software repository from which to extract the source code artifacts can inject important biases in the results of our evaluation. We have expanded our discussion of this aspect in the threats to validity section of the manuscript, as in the following:

*"All considered source files were small programs collected from a single software repository. The said software repository itself was implemented for a specific purpose, namely the evaluation of the performance of different programming languages at runtime. Therefore, it is still unsure whether our measurements can scale up to bigger software repositories and real-world applications written in the evaluated languages. As well, the results of the present manuscript may inherit possible biases that the authors of the code had in writing the source artifacts employed for our evaluation. Future extensions of the current work should include the computation of the selected metrics on more extensive and more diverse sets of software artifacts to increase the generalizability of the present results."*

- With regards to RQ1, authors did not detail the relation between code verbosity and maintainability. Existing methodologies to determine maintainability, and at a higher level than the MI, such as technical debt are concerned with existing best practices, detection of code smells and other weaknesses; as such, it is unclear how the innate verbosity of a language will translate to the maintainability characteristic.

We have added a paragraph at the end of the discussion of the RQ1 where we briefly discuss why verbosity can be considered as a primary proxy for readability and understandability. We report the paragraph below:

*Albeit many higher-level measures and metrics have been derived in the latest years by related literature to evaluate the understandability and maintainability of software, the analysis of code verbosity can be considered a primary proxy for these evaluations. Several studies, in fact, have linked the intrinsic verbosity of a language to lower readability of the software code, which translates to higher effort when the code has to be maintained. For instance, Flauzino et al., state that verbosity can cause higher mental energy in coders working on the implementation of an algorithm, and can be correlated to many smells in software code. Toomim et al. highlight that redundancy and verbosity can obscure meaningful information in the code, thereby making it difficult to understand.*

- Regarding the authors' answer to RQ2, the discussion should be based on the implementation of larger-scale software; it should also include a discussion on the source

code author(s) programming style, as that can have an impact on these complexity metrics, especially when considering such a limited code base. This is true especially in the case of the NARGS and NEXITS metrics that are not extensively studied in the literature.

We have partly addressed this comment in responses to previous comments by the same reviewer. We report in the following the fundamental points:

- Regarding large-scale software: we have emphasized in the threats to validity issue the fact that it cannot be guaranteed that the results of our evaluation can scale up to bigger projects;

- we have added in the Threats to validity section that the programming style of the authors of the source artifacts can largely impact the metrics, and that this impact would be amplified by the small size of the code based employed for the experiment.

- we have added a threat related to the limited validity of the NARGS and NEXITS metrics and in the discussion of the metric we have toned down all the statements linking NARGS and NEXITS to maintainability, given that no findings exist in the literature that demonstrates the correlation between these metrics and such property of source code.

- The application of the Halstead time and bugs metrics to a new programming language/construct introduces further threats to validity; these proposed values (division by 18 and 3000, respectively) should most likely be evaluated empirically first. This is partly addressed by the authors in the Threats to Validity section.

As pointed out by the reviewer, the selection of the specific Halstead coefficients may largely impact the final metric computations. An empirical evaluation of the parameters for the Rust language would be needed, but unfortunately is not feasible in the time and size of the current research and manuscript and can only be considered as future work.

We have however extended the paragraph in the threats to validity section that tackles the limitation of using standard coefficients, by pointing out which metrics can be impacted and what would be needed to enhance the validity of the findings. We report the paragraph below:

*The values measured for the individual metrics (and, by consequence, the reasoning based upon them) can be heavily influenced by the exact formula used for the metric computation. In the Halstead suite, the formulas depend on two coefficients defined explicitly in the literature for every software language, namely the denominators for the T and B metrics. Since no previous result in the literature has provided Halstead coefficients specific to Rust, we used the C coefficients for the computation of Rust Halstead metrics. More specifically, we used 18 as the denominator of the T metric. This value, called Stoud number (S), is measured in moments, i.e. the time required by the human brain to carry out the most elementary decision. In general, S is comprised between 5 and 20. In the original Halstead metrics suite for the C*

*language, a value of 18 is used. This value was empirically defined after psychological studies of the mental effort required by coding. We selected 3000 as the denominator of the Number of delivered Bugs metric; this value, again, is the original value defined for the Halstead suite and represents the number of mental discriminations required to produce an error in any language. The 3000 value was originally computed for the English language and then mutuated for programming languages.*

*The choice of the Halstead parameters may significantly influence the values obtained for the T and B metrics. The definition of the specific parameters for a new programming language, however, implies the need for a thorough empirical evaluation of such parameters. Future extensions of this work may include studies to infer the optimal Halstead parameters for Rust source code.*

- I am not convinced that RQ1 - RQ3 are related to software maintainability, as it is understood from a software engineering perspective.
The paper is competently written and approaches a subject of current interest in research. However, I believe that the title is out of sync with the paper's contents. The selection of target applications is severely limited, and suitable for an introductory, or position paper on the subject, and not a full journal publication. Furthermore, the selection of metrics to assess maintainability is limited to simplistic measurements. Recent research into maintainability generally employs more complex measures such as technical debt or the impact of code smells (such as measured using SonarQube or Ptidej). Of course, their application assumes a larger target code base to provide meaningful results. Relating to the selection of target applications, the 9 implementations are part of a benchmarking suite, and as such introduce an important threat related to the validity of the conclusions, when these are externalized to other kinds of software (e.g. open-source world or proprietary implementations of large-scale systems).

I believe that in order to work well, the paper should be re-targeted towards examining source code verbosity and understandability across different languages. In this way, the selection of target applications gains relevance, and well-known metrics such as the Halstead suite (that are no longer used to evaluate maintainability) can be more successfully employed.

Verbosity, complexity, and code organization have been considered - in many categorizations of software static metrics - as proxies for code maintainability. We indeed agree that finer ways to measure or estimate the maintainability of software projects are available in both literature and practice. Following the reviewer's suggestion, we have toned down the claims about Rust maintainability throughout all of the paper, and we have changed the paper by mentioning explicitly which static measurements have been actually carried out in our work.

Therefore, the new title of the manuscript is "Evaluation of Rust Code Verbosity, Understandability and Complexity".

Significant modifications have been applied to the Introduction, Background, Discussion, and Conclusion sections of the manuscript.

**Reviewer 3**

Authors have followed the professional article structure and shared the raw data. I commend the authors for their work but certain issues need to be resolved before acceptance.

1. Mapping of figures/tables should be thoroughly cross checked with the places they are referenced in the manuscript. Authors need to correct the table and figure referencing
For example,.
In 358, "In the table, we report the mean and......." Which table authors are referring to?
Line 403 "The boxplots in Figure 4 and Table 9 report the distributions, mean, and median of the Halstead...." Authors have cited wrong table and figure reference. It should be Figure 5 and Table 10.
Similar observation is made at line 435-436. "The boxplots in Figure 4 and Table 9 report the distributions, mean, and median of the Maintainability Indexes computed for the six different programming languages.". This is repeated line with wrong references.

> We have corrected the wrong references and numberings highlighted by the reviewer, along with others that came from an additional proof-read of the manuscript

2. he authors use "we" too much in the paper, while I suggest to use "the paper".

> We recognize that we have used the personal form too many times throughout the manuscript. We have substituted it in the paper with usages of the passive form and with different subjects (the paper, the section, the table)

3. Different notations are used for same object. It is recommended that authors should use single term. Some examples are;
Json, json, .json;
line 321 SLOC line 322 souce loc. In line 323 and 324 lines of code. C OGNITIVE complexity (line 375, 376) or Cognitive complexity (line 390, 395, 396), or cognitive complexity (line 393) or Cognitive Complexity(396).
Program difficulty ( line 406, 409) and Difficulty(line 408)

> We have uniformed all the different spellings to JSON, SLOC, COGNITIVE

4. As authors has mentioned in line 438, "Halstead Volume (V), the Cyclomatic Complexity (CC),...", they must mention the acronyms for all other terms when first used in paper.

> We have added the clarification of the acronyms every time we cite a metric for the first time in the paper

5. The paper is well organized. But at some points, restructuring of sentences is required. Few examples are:

Line 269-271: Multiple use of and in one sentence. "Concerning the original implementation of the rust-code-analysis tool, we have forked the project and performed modifications on it by adding metrics computations (e.g., the COGNITIVE metric) and changes to the possible output format provided by the tool."

Line 440-441: "By using all the formulas for the Maintainability Index, we computed for the source files written in Rust an average MI that placed the fourth among all considered programming languages."
"This very low value of the cognitive per method for Rust is related…….." should be "This very low value of the cognitive complexity per method for Rust is related…….."

6. Minor grammatical errors were exposed. For example,
In 335 "….with the second-highest, mean being 59 for the….." should be "….with the second-highest mean being 59 for the….."

In captions of Figure 1, Figure 2 :" Distributions of the metrics about….." should be "Distribution of the metrics about…".
Line 95: Systematic Literature review should be systematic literature review.

7. Please check the PeerJ reference format and references should be consistent with that format.
The references in manuscript do not follow a commat format. For example,
Alqadi, B. S. and Maletic, J. I. (2020). Slice-based cognitive complexity metrics for defect prediction. In 2020 IEEE 27th International Conference on Software Analysis, Evolution and Reengineering (SANER), pages 411–422. IEEE.
Astrauskas, V., Mu¨ller, P., Poli, F., and Summers, A. J. (2019). Leveraging rust types for modular specification and verification. Proceedings of the ACM on Programming Languages, 3(OOPSLA):1– 30.

> All required fixes were performed

Experimental design
The experiments were well implemented, and the results are consistent. Work is novel. A tool is constructed to extract metrics of Rust and object-oriented languages. Metrics are collected for 9 program codes written in 6 programming languages. The paper is well written, the structure makes it easy to follow. Research questions are well formulated.

1. I would request authors to comment on their selection of metrics to be extracted from code. Why they did not extract object-oriented metrics?

> Rust-code-analyis computes metrics on both functions and class methods. Currently rust-code-analysis does not implement any class metric, so they haven't been considered in our analysis. We consider to increase the number of metrics of the rca tool in our future work, and we have now made it explicit in the Conclusion and Future Work section.

*As the prosecution of this work, we plan to perform further developments on the rust-code-analysis tool such that it can provide more metric computation features. At the present time, for instance, the tool is not capable of computing class-level and object-oriented metrics, but it can only be employed to compute metrics only on function and class methods.*

2. Algorithms are language independent. Authors have use codes of different languages to do comparative analysis. In Table 6 title, algorithms should be replaced by code. Similarly, in complete text, whenever referring to code, replace 'algorithm' by 'code'.

We removed the ambiguity without "algorithm" and "code" (i.e., the implementation of the algorithm in a specific language) throughout the paper and in the mentioned table.

3. Authors mentioned and analyzed maintainability index in subsection 4.4. I would suggest authors to include some range of maintainability index (for example: bad, average, good, acceptable). This will give more clarity to readers about its relevance.

The ranges for bad, medium, and good maintainability of the source code have been reported in table 6 and briefly discussed:

*For the traditional and the SEI formulas of the MI, a value over 85 indicates easily maintainable code; a value between 65 and 85 indicates average maintainability for the analyzed code; a value under 65 indicates hardly maintainable code. With the original and SEI formulas, the MI value can also be negative. With the Visual Studio formula, the thresholds for medium and high maintainability are moved respectively to 10 and 20.*

4. In Table 5, authors have scribed the three variants of MI metric. It is suggested to add reference and little detail for each definition in corresponding section.

We have added descriptions and references for each of the formulas. We report in the following the excerpt from the paper:

*To answer RQ4, the Maintainability Index was adopted, i.e., a composite metric originally defined by Oman et al. to provide a single index of maintainability for software [34].*

*Three different versions of the Maintainability Index are considered. First, the original version by Oman et al.. Secondly, the version defined by the Software Engineering Institute (SEI), originally promoted in the C4 Software Technology Reference Guide [35]; the SEI adds to the original formula a specific treatment for the comments in the source code (i.e., the CLOC metric), and it is deemed by research as more appropriate given that the comments in the source code can be considered correct and appropriate [35]. Finally, the version of the MI metric implemented in the Visual*

*Studio IDE [36]; this formula resettles the MI value in the 0-100 range, without taking into account the distinction between CLOC and SLOC operated by the SEI formula [37].*

*The respective formulas are reported in Table 5. The interpretation of the measured MI varies according to the adopted formula to compute it: the ranges for each of them are reported in Table 6.*

5. Table 1 shows that CKJM extracts JAVA and C metrics.
But CKJM collects metrics only for compiled JAVA classes. CKJM stands for Chidamber and Kemerer Java Metrics. It does not work for C code. Authors need to rectify it.
Spinellis D. Tool writing: a forgotten art?(software tools). IEEE Software. 2005 Jul 11;22(4):9-11
Validity of the findings
I appreciate authors to provide all underlying data supporting the replication of the work.

We have fixed the table.

1. In results section, conclusions are well stated for each RQ. But comparative analysis need to be further strengthened by using statistical tests. Authors must include statistical validation of their results. Depending on the nature of data, they can use either parametric or non-parametric tests to statistically validate the results.

We have applied a non-parametric Kruskal-Wallis test to identify statistically significant differences among the different sets of metrics for each language.

For significantly different distributions we have finally applied post-hoc comparisons with Wilcoxon signed rank sum test to analyze the difference between the metrics measured for Rust and the other five languages in the set.

2. Conclusion section need to be elaborated. Authors should include main contributions in it.

We have added the main findings and contributions of the manuscript in the conclusion section.

---

## Round 0.3 · accepted · Accept

The manuscript has been revised and has been accepted.